# Duox-generated reactive oxygen species activate ATR/Chk1 to induce G2 arrest in *Drosophila* tracheoblasts

**Amrutha Kizhedathu[1], Piyush Chhajed[1], Lahari Yeramala[2], Deblina Sain Basu[1,3], Tina Mukherjee[1], Kutti R Vinothkumar[2], Arjun Guha[1]***

[1]Regulation of Cell Fate, Institute for Stem Cell Science and Regenerative Medicine (inStem), Bangalore, India; [2]National Centre for Biological Sciences, Tata Institute of Fundamental Research, Bangalore, India; [3]Trans Disciplinary University, Bangalore, India

**Abstract** Progenitors of the thoracic tracheal system of adult *Drosophila* (tracheoblasts) arrest in G2 during larval life and rekindle a mitotic program subsequently. G2 arrest is dependent on ataxia telangiectasia mutated and rad3-related kinase (ATR)-dependent phosphorylation of checkpoint kinase 1 (Chk1) that is actuated in the absence of detectable DNA damage. We are interested in the mechanisms that activate ATR/Chk1 (Kizhedathu et al., 2018; Kizhedathu et al., 2020). Here we report that levels of reactive oxygen species (ROS) are high in arrested tracheoblasts and decrease upon mitotic re-entry. High ROS is dependent on expression of Duox, an $H_2O_2$ generating dual oxidase. ROS quenching by overexpression of superoxide dismutase 1, or by knockdown of Duox, abolishes Chk1 phosphorylation and results in precocious proliferation. Tracheae deficient in Duox, or deficient in both Duox and regulators of DNA damage-dependent ATR/Chk1 activation (ATRIP/TOPBP1/claspin), can induce phosphorylation of Chk1 in response to micromolar concentrations of $H_2O_2$ in minutes. The findings presented reveal that $H_2O_2$ activates ATR/Chk1 in tracheoblasts by a non-canonical, potentially direct, mechanism.

**\*For correspondence:**
arjung@instem.res.in

**Competing interest:** The authors declare that no competing interests exist.

## Introduction

Ataxia telangiectasia mutated and rad3-related kinase (ATR, *Mei-41*) and its substrate, checkpoint kinase 1 (Chk1, *Grapes*), are essential for DNA damage repair and for normal development (*Artus and Cohen-Tannoudji, 2008*; *Blythe and Wieschaus, 2015*; *Cimprich and Cortez, 2008*). The mechanism by which ATR phosphorylates and activates Chk1 in response to DNA damage and the mechanism by which activated Chk1 induces cell cycle arrest have been well characterized (*Choi et al., 2010*; *Cimprich and Cortez, 2008*; *Delacroix et al., 2007*; *Lee et al., 2012*; *Xu and Leffak, 2010*). In contrast, the mechanisms for the activation of ATR/Chk1 during development, and the roles of these proteins therein, are less well understood. We reported recently that the ATR/Chk1 axis is co-opted during *Drosophila* development for inducing G2 arrest in progenitor cells (*Kizhedathu et al., 2018*; *Kizhedathu et al., 2020*). The current study was designed to shed light on the mechanism by which the pathway is activated in this context and to probe if it is distinct from the mechanism for DNA damage-induced activation.

The progenitors of the adult tracheal (respiratory) system in *Drosophila* remain mitotically quiescent through larval life and rekindle cell divisions at the onset of pupariation. Progenitors of the tracheal branches of the second thoracic metamere (Tr2, hereafter referred to as tracheoblasts) are arrested in the G2 phase of the cell cycle for ~56 hr and divide thereafter (*Djabrayan et al., 2014*). Importantly, we have shown previously that G2 arrest in tracheoblasts is dependent on ATR-dependent

phosphorylation of Chk1. Two aspects of this process are striking and merit mention here. First, arrested tracheoblasts phosphorylate Chk1 (phosphorylated Chk1 [pChk1]) in the absence of detectable DNA damage. Second, Chk1 is upregulated by a Wnt signaling-dependent mechanism in arrested tracheoblasts and high levels of Chk1 expression are necessary to induce arrest (*Kizhedathu et al., 2018*; *Kizhedathu et al., 2020*). In the absence of detectable DNA damage, the mechanism for the activation of ATR/Chk1 pathway in tracheoblasts has been unclear from the studies so far.

Although ATR and the related kinases ataxia telangiectasia mutated (ATM) and DNA-PK are the principal sensors of DNA damage and effectors of the DNA damage response (*Durocher and Jackson, 2001*), there is evidence that these kinases can also be activated by non-canonical mechanisms that are not dependent on DNA damage (*Guo et al., 2010*; *Kumar et al., 2014*). Reactive oxygen species (ROS) are a group of oxygen-derived small molecules that interact avidly with macromolecules like proteins, lipids, and nucleic acids, and alter their function (*Corcoran and Cotter, 2013*). Pertinently, there is evidence that ROS can directly activate ATM by stabilizing ATM homodimers through the formation of intermolecular disulfide bridges (*Guo et al., 2010*). ROS-activated ATM can phosphorylate and activate substrates like checkpoint kinase 2 (Chk2). Whether ROS can activate ATR in the same manner is not known. There is evidence that ROS can activate ATR and lead to Chk1 phosphorylation via the induction of DNA damage (*Srinivas et al., 2019*; *Willis et al., 2013*). A recent study has shown that mechanical stress on the nuclear envelope, arising during DNA replication (S phase) or in response to osmotic stress or mechanical stimulation, can lead to ATR activation (*Kumar et al., 2014*). This implies that ATR can also be activated by a non-canonical mechanism.

ROS are now generally recognized as important regulators of developmental processes (*Zhang et al., 2016*). The regulation of ROS levels in cells during development is orchestrated by altering rates of aerobic respiration in cells or by altering the levels of expression of ROS-producing enzymes like NADPH oxidases (NOXs). Interestingly, the *Drosophila* genome encodes two NADPH oxidases – NOX and dual oxidase (Duox) (*Kim and Lee, 2014*) – and one of these, Duox, is expressed at high levels in the larval tracheal system (*Robinson et al., 2013* ). Duox catalyzes the oxidation of NADPH, leading to the generation of $H_2O_2$ (*Bedard and Krause, 2007*; *Geiszt et al., 2003*).

Our interest in the mechanism for activation of ATR/Chk1 led us to probe the role of ROS in this context. We found that ROS levels are high when the cells are arrested in G2 and low after the cells rekindle cell division. We quenched ROS in tracheoblasts via overexpression of the superoxide dismutase 1 (SOD1) to find that the cells started dividing precociously in a manner similar to Chk1 mutants. These findings led us to probe the role of Duox in the regulation of ROS and the mechanism by which ROS regulate the ATR/Chk1 axis.

## Results

### High ROS is required for G2 arrest in larval tracheoblasts

The cells that comprise the tracheal branches of the second thoracic metamere (Tr2) of the larvae are differentiated adult progenitors that contribute to the development of pupal and adult tracheal structures (*Djabrayan et al., 2014*; *Guha et al., 2008*; *Guha and Kornberg, 2005*). The cells that make up the dorsal trunk (DT) in Tr2, hereafter referred to as tracheoblasts, are the focus of our studies. Tracheoblasts remain arrested in the G2 phase of the cell cycle during larval life and initiate mitosis thereafter. Analysis of the cell cycle phasing of tracheoblasts using the fluorescent ubiquitination-based cell cycle indicator (FUCCI) system (*Zielke et al., 2014*) has shown that the cells are in the G1 phase at the time the embryo hatches into a larva and that the cells transition from G1 to S to G2 in the first larval instar (L1). Tracheoblasts remain in G2 from the second larval instar (L2) till mid third larval instar (L3) (32–40 hr L3, ~56 hr) and divide rapidly thereafter (*Kizhedathu et al., 2018*; *Kizhedathu et al., 2020*, *Figure 1A*).

To probe the role of ROS in the regulation of G2 arrest in tracheoblasts, we assessed the levels of cytoplasmic ROS in tracheoblasts at L2, 0-8 hr L3, 16-24 hr L3 and 32–40 hr L3 using two well-established, redox-sensitive dyes: 2′,7′-dichlorodihydrofluorescein diacetate (H₂DCFDA) and dihydroethidium (DHE). Both H₂DCFDA and DHE are cell-permeable molecules that alter light emission upon oxidation (*Yang et al., 2014*). Analysis of H₂DCFDA and DHE staining in tracheoblasts at various stages revealed that levels of both reporters are readily detectable at L2 (*Figure 1B and F*, *Figure 1—figure supplement 1A and B*, n ≥ 6 tracheae per condition per experiment, n = 3), 0–8 hr L3 (*Figure 1C*

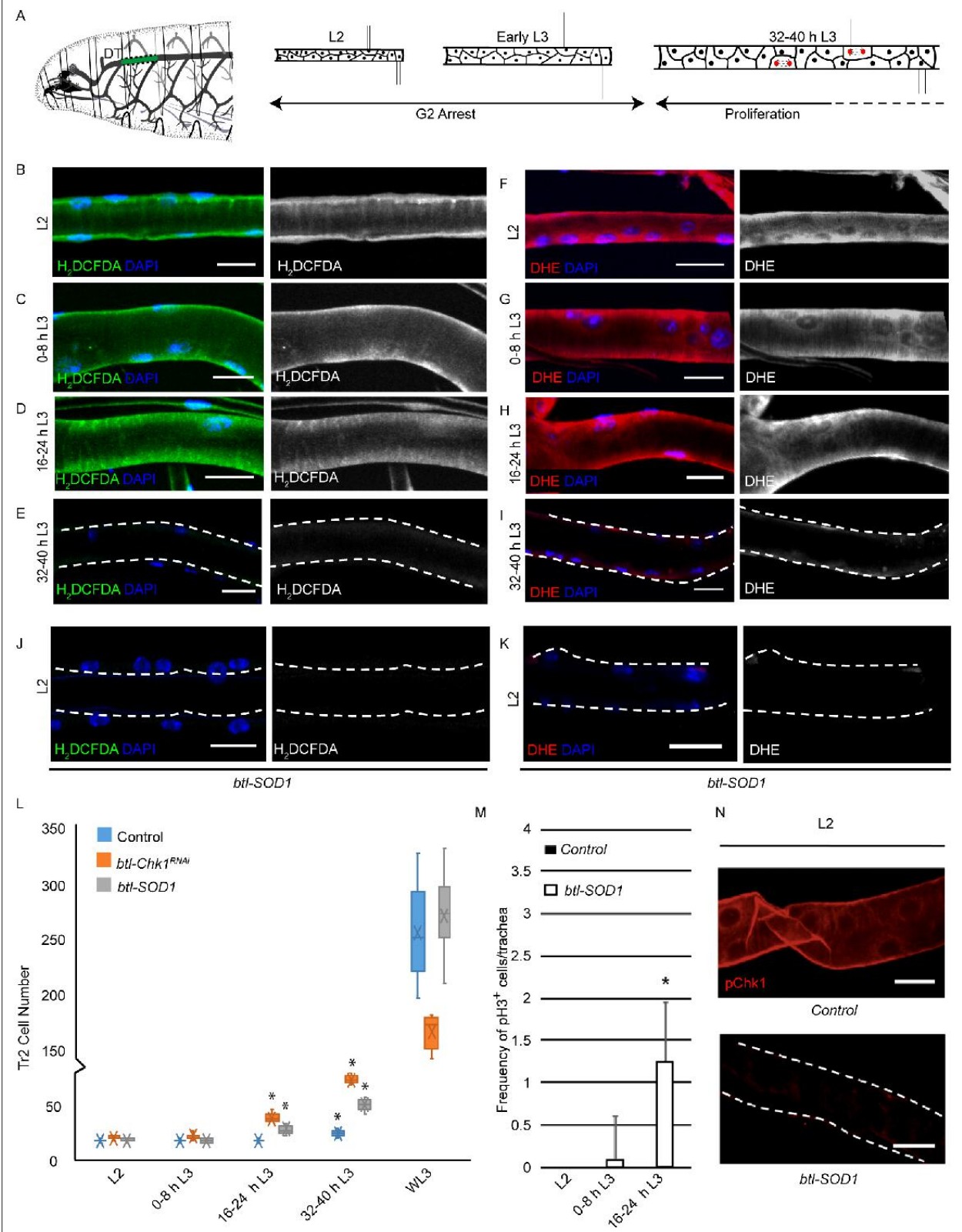

**Figure 1.** High levels of reactive oxygen species (ROS) are required for checkpoint kinase 1 (Chk1) activation and G2 arrest in tracheoblasts. (**A**) A diagram of the third instar larva showing the dorsal trunk (DT) of the second thoracic metamere (Tr2, colored in green and marked by dashed line). The diagram also shows the timecourse of G2 arrest and cell division in Tr2. The cells in Tr2 DT remain geographically isolated from tracheal cells in other branches during larval life. (**B–E**) Levels of the ROS reporter 2',7'-dichlorodihydrofluorescein diacetate (H$_2$DCFDA) in Tr2 DT during larval stages.

*Figure 1 continued on next page*

*Figure 1 continued*

Shown in the figures are $H_2DCFDA$ staining in L2 (**B**), 0–8 hr L3 (**C**), 16–24 hr L3 (**D**), and 32–40 hr L3 (**E**) in wild type (*btl-Gal4*) animals. (**F–I**) Levels of the ROS reporter dihydroethidium (DHE) in Tr2 DT during larval stages. Shown in the figures are DHE staining in L2 (**F**), 0–8 hr L3 (**G**), 16–24 hr L3 (**H**), and 32–40 hr L3 (**I**) in wild type (*btl-Gal4*) animals. (**J, K**) Effect of *btl*-Gal4-dependent overexpression of superoxide dismutase 1 (SOD1) on levels of ROS reporters in Tr2 DT. (**J**) $H_2DCFDA$ staining in *btl-SOD1* (*btl-GAL4/UAS-SOD1*)-expressing larvae (n ≥ 6 tracheae per condition per timepoint). (**K**) DHE staining in *btl-SOD1* larvae (n ≥ 6 tracheae per condition per timepoint). (**L**) Effect of SOD1 overexpression on cell numbers in Tr2 DT at different larval stages. Graph shows numbers of Tr2 tracheoblasts in wild type (*btl-Gal4*), *btl-SOD1* (*btl-GAL4/UAS-SOD1*), and *btl-Chk1^{RNAi}* (*btl-GAL4/UAS-Chk1^{RNAi}*) larvae at L2, 0–8 hr L3, 16–24 hr L3, 32–40 hr L3, and wandering L3 (WL3) (n ≥ 7 tracheae per condition per timepoint). (**M**) Effect of SOD1 overexpression on mitotic indices in Tr2 DT (see text). Graph shows mitotic indices in Tr2 DT in wild type and *btl-SOD1* (*btl-GAL4/UAS-SOD1*)-expressing larvae at L2, 0–8 hr L3 and 16–24 hr L3 (mean values ± standard deviation, n ≥ 7 tracheae per condition per timepoint). (**N**) Effect of SOD1 overexpression on Chk1 phosphorylation in Tr2 tracheoblasts. Shown in the figure is phosphorylated Chk1 (pChk1, phospho-Chk1Ser^{345}) immunostaining (red) in Tr2 DT in wild type (*btl-GAL4*) and *btl-SOD1* (*btl-GAL4/UAS-SOD1*) larvae at L2. Scale bars = 10 µm. Dashed lines outline the cuticular lumen of the tracheal tube here and elsewhere and are shifted outward to include the epithelial lining when they overlap with the signal. Student's t-test: *p<0.00001.

The online version of this article includes the following figure supplement(s) for figure 1:

**Source data 1.** Cell Frequencies in SOD1 overexpressing animals.

**Source data 2.** Mitotic indices in SOD1 overexpressing animals.

**Figure supplement 1.** Quantification of reactive oxygen species (ROS) levels in Tr2 tracheoblasts.

**Figure supplement 2.** Quantification of phosphorylated checkpoint kinase 1 (pChk1) levels in Tr2 tracheoblasts.

and G, *Figure 1—figure supplement 1A and B*, n ≥ 6 tracheae per condition per experiment, n = 3), 16–24 hr L3 (*Figure 1D and H*, *Figure 1—figure supplement 1A, B*, n ≥ 6 tracheae per condition per experiment, n = 3), and nearly undetectable at 32–40 hr L3 in wild type (*btl-Gal4*) animals (*Figure 1E, I*, *Figure 1—figure supplement 1A and B*, n ≥ 6 tracheae per condition per experiment, n = 3). Taken together, the analysis of ROS reporters (*Figure 1*) showed that cytoplasmic ROS is high in arrested cells and low in mitotically active cells.

Next, we asked whether changes in ROS levels had any bearing on the cell cycle program. To answer this question, we adopted a genetic approach for quenching ROS in tracheoblasts. SOD1 is a cytoplasmic enzyme that scavenges ROS (*Blackney et al., 2014*). We overexpressed SOD1 in trachea (*btl-GAL4/UAS-SOD1*, hereafter *btl-SOD1*) and examined the effects of SOD1 overexpression on ROS levels by $H_2DCFDA$ and DHE staining. Levels of $H_2DCFDA$ and DHE were found to be significantly lower in *btl-SOD1*-expressing animals compared to controls (*Figure 1J and K*, *Figure 1—figure supplement 1C and D*, compare with *Figure 1B and F*, n ≥ 6 tracheae per condition per experiment, n = 3). This showed that SOD1 overexpression is an effective way to quench ROS in tracheae. We then determined whether SOD1 overexpression altered the cell cycle program of tracheoblasts. We counted the number of cells in Tr2 DT at L2, 0–8 hr L3, 16–24 hr L3, 32–40 hr L3, and wandering L3 (WL3) and quantified the frequencies of phospho-histone H3$^+$ (pH3$^+$) mitotic figures in Tr2 DT at L2, 0–8 hr L3, and 16–24 hr L3. Analysis of the cell numbers and frequency of pH3$^+$ figures in Tr2 showed that SOD1 overexpression resulted in precocious cell division from 0 to 8 hr L3 (*Figure 1L and M*, n ≥ 7 tracheae per timepoint, *Figure 1—source data 1*; *Figure 1—source data 2*). We noted that the precocious cell divisions in *btl-SOD1* tracheae are similar to that observed in Chk1 mutants (*btl-GAL4/UAS-Chk1^{RNAi}* (*btl*-Chk1^{RNAi}*), *Figure 1L*, *Figure 1—source data 1*). Based on these data, we concluded that quenching of ROS phenocopies the loss of Chk1 in these cells. We also noted a difference in cell proliferation kinetics in *btl-Chk1^{RNAi}* and *btl-SOD1*-expressing animals. Tracheoblasts in *btl-Chk1^{RNAi}* rekindle mitoses earlier than wild type but divide more slowly thereafter. In contrast, tracheoblasts in *btl-SOD1* rekindle mitoses earlier than wild type but do not appear to divide more slowly than wild type cells (see Discussion).

The findings above led us to investigate the levels of phosphorylated (activated) Chk1 in tracheoblasts in wild type and *btl-SOD1* animals. As reported previously, pChk1 levels are high in L2 and early L3 and diminished at 32–40 hr L3. pChk1 immunostaining in *btl-SOD1*-expressing tracheae in L2 and early L3 showed that pChk1 levels were reduced in comparison to wild type at these respective stages (*Figure 1N*, *Figure 1—figure supplement 2A*, n ≥ 6 tracheae per condition per experiment, n = 3). We inferred that high ROS levels are necessary for G2 arrest and that high ROS contributes in some manner to high levels of pChk1.

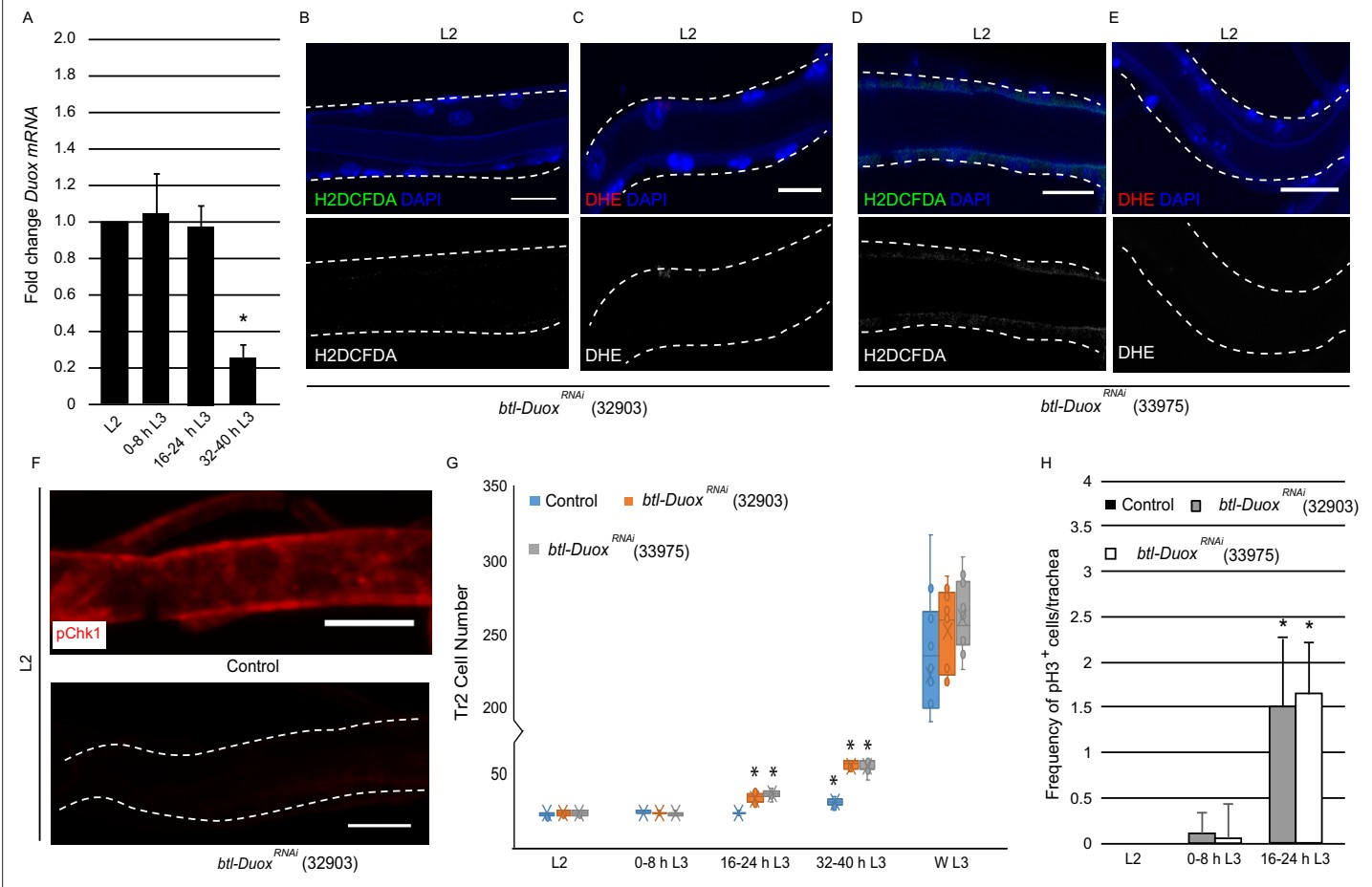

**Figure 2.** High reactive oxygen species (ROS) in tracheoblasts is dependent on *Duox* expression. (**A**) Quantitative PCR analysis of *Duox* mRNA levels in micro-dissected Tr2 dorsal trunk (DT) fragments at different stages. Graph shows fold change in *Duox* mRNA in Tr2 DT fragments from wild type (*btl-GAL4*) larvae at L2, 0–8 hr L3, 16–24 hr L3, and 32–40 hr L3. Fold change has been represented with respect to L2 (n = 3 experiments, n ≥ 15 Tr2 DT fragments/stage/experiment, mean ± standard deviation, p<0.0001). (**B–E**) Effect of the knockdown of *Duox* expression on the levels of ROS reporters in Tr2 DT in L2. Shown here are the results of the expression of two different Duox RNAi lines (32903 and 33975). (**B, D**) 2',7'-Dichlorodihydrofluoresce in diacetate (H₂DCFDA) staining and (**C, E**) dihydroethidium (DHE) staining in Tr2 DT in *btl-Duox*^RNAi^. (**[B, C]** *btl-GAL4/+; UAS-Duox*^RNAi^ *(32903)/+* and **[D, E]** *btl-GAL4/+; UAS-Duox*^RNAi^ *(33975)/+*) larvae (n ≥ 6 tracheae per condition per timepoint). (**F**) Effect of reduction of *Duox* expression on levels of phosphorylated checkpoint kinase 1 (pChk1) in Tr2 DT in L2. pChk1 immunostaining (red) in Tr2 DT in wild type (*btl-Gal4*) and *btl-Duox*^RNAi^ (*btl-GAL4/+; UAS- Duox*^RNAi^ *(32903)/+*) larvae. (**G**) Effect of the knockdown of *Duox* expression on cell numbers in Tr2 DT at different larval stages. Graph shows cell numbers of Tr2 tracheoblasts in wild type (*btl-Gal4*) and *btl-Duox*^RNAi^ (*btl-GAL4/+; UAS-Duox*^RNAi^ *(32903)/+* and *btl-GAL4/+; UAS-Duox*^RNAi^ *(33975)/+*) larvae at L2, 0–8 hr L3, 16–24 hr L3, 32–40 hr L3, and WL3 (n ≥ 7 tracheae per condition per timepoint). (**H**) Effect of the knockdown of *Duox* expression on mitotic indices in Tr2 DT. Graph shows mitotic indices in Tr2 DT in wild type (*btl-Gal4*) and *btl-Duox*^RNAi^ (*btl-GAL4/+; UAS-Duox*^RNAi^ *(32903)/+* and *btl-GAL4/+; UAS-Duox*^RNAi^ *(33975)/+*) larvae at L2, 0–8 hr L3 and 16–24 hr L3 (mean values ± standard deviation, n ≥ 7 tracheae per condition per timepoint). Scale bars = 10 µm. Student's t-test: *p<0.0001

The online version of this article includes the following figure supplement(s) for figure 2:

**Source data 1.** Cell frequencies in Duox ^RNAi^ expressing animals.

**Source data 2.** Mitotic indices in Duox ^RNAi^ expressing animals.

## High ROS in G2-arrested tracheoblasts is dependent on *Duox*

The identification of ROS as regulator of G2 arrest in tracheoblasts raises two questions. First, how are ROS levels regulated in the tracheoblasts? Second, how does high ROS translate into high levels of pChk1? As mentioned previously, the H₂O₂-generating enzyme Duox is expressed at high levels in larval tracheae (*Robinson et al., 2013*). To probe the role of Duox in the generation of ROS in tracheoblasts, we first examined Duox expression in tracheoblasts at different larval stages using quantitative RT-PCR (qPCR). We isolated mRNA from micro-dissected fragments of Tr2 DT at different timepoints

and utilized these samples to query Duox expression. Our analysis showed that Duox mRNA levels are higher at L2, 0–8 hr L3, and 16–24 hr L3 than at 32–40 hr L3 (**Figure 2A,** n ≥ 15 tracheal fragments per timepoint per experiment, n = 3 experiments). We concluded that the timecourse of Duox mRNA expression correlates with the timecourse of ROS accumulation in the tracheae.

To probe whether Duox is the driver of ROS accumulation, we knocked down the levels of Duox in the tracheal system by RNA interference and examined its impact on the levels of ROS reporters. The reduction in the levels of Duox using two different RNAi lines (BDSC-32903, BDSC-33975, *btl-GAL4/+; UAS-Duox^{RNAi}/+ (btl-Duox^{RNAi})*) followed by $H_2$DCFDA and DHE staining showed that the reduction of Duox leads to a dramatic decrease in levels of both the reporters (**Figure 2B–E**, **Figure 1—figure supplement 1E and F**, compare with **Figure 1B and E**, n ≥ 6 tracheae per condition per experiment, n = 3). Based on these data, we inferred that the high levels of ROS in arrested tracheoblasts are dependent on Duox expression.

Next, we examined how the knockdown of Duox impacted Chk1 phosphorylation and the program of cell division. Consistent with the previous findings with SOD1 overexpression, we observed that the knockdown of Duox resulted in the loss of pChk1 (**Figure 2F**, **Figure 1—figure supplement 2B**, n ≥ 6 tracheae per condition per experiment, n = 3). We counted the number of cells of tracheoblasts at L2, 0–8 hr L3, 16–24 hr L3, 32–40 hr L3, and WL3 and quantified the frequencies of pH3+ nuclei in Tr2 DT at L2, 0–8 hr L3, and 16–24 hr L3. We found that *btl-Duox^{RNAi}*-expressing animals rekindle cell divisions sooner than their wild type counterparts. Interestingly, *btl-Duox^{RNAi}*-expressing animals also showed

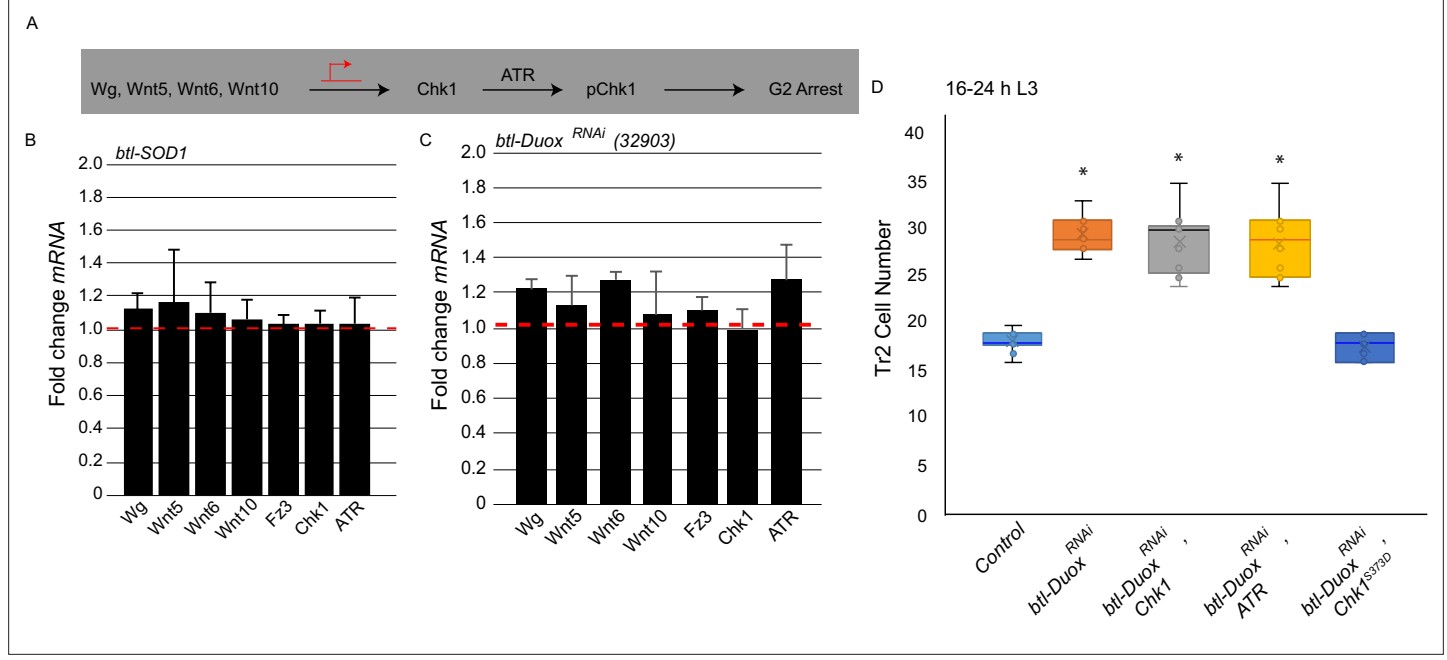

**Figure 3.** Reactive oxygen species (ROS) dependence identifies a novel pathway for the regulation of ataxia telangiectasia mutated-related kinase/checkpoint kinase 1 (ATR/Chk1) in tracheoblasts. (**A**) Model for G2 arrest mechanism in Tr2 tracheoblasts based on previous studies. Earlier work has shown that four Wnt ligands (*Wg, Wnt5, Wnt6, Wnt10*) act synergistically to upregulate Chk1 mRNA levels in arrested tracheoblasts. High levels of Chk1 expression are necessary for G2 arrest and Chk1 overexpression can rescue defects in Wnt signaling (**Kizhedathu et al., 2020**). (**B, C**) Effect of superoxide dismutase 1 (SOD1) overexpression and Dual oxidase (Duox) knockdown on expression of Wnts and Wnt-target genes. Quantitative PCR analysis of levels of *Wg, Wnt5, Wnt6, Wnt10, Fz3, Chk1,* and *ATR* mRNA in micro-dissected Tr2 DT fragments at L2. Graph shows fold change in *Wg, Wnt5, Wnt6, Wnt10, Fz3, Chk1,* and *ATR* mRNA levels in Tr2 dorsal trunk (DT) fragments expressing (**B**) *btl-SOD1 (btl-GAL4/UAS-SOD1)* and (**C**) *btl-Duox^{RNAi} (btl-GAL4/+; UAS-Duox^{RNAi} (32903)/+)*. Fold change has been represented with respect to wild type (*btl-Gal4,* shown by dashed red line, n = 3 experiments, n ≥ 15 Tr2 DT fragments/stage/experiment, mean ± standard deviation). (**D**) Effect of overexpression of a phosphomimic variant of Chk1 in *btl-Duox^{RNAi}* larvae at 16–24 hr L3. Graph shows numbers of Tr2 tracheoblasts in wild type (*btl-Gal4*), *btl-Duox^{RNAi} (btl-GAL4/+; UAS-Duox^{RNAi} (32903)/+)*, *btl-Duox^{RNAi}, Chk1 (btl-GAL4/+; UAS-Duox^{RNAi} (32903)/ UAS-Chk1), btl-Duox^{RNAi}, ATR (btl-GAL4/+; UAS-Duox^{RNAi} (32903)/ UAS-ATR)* and *btl-Duox^{RNAi}, Chk1^{S373D} (btl-GAL4/+; UAS-Duox^{RNAi} (32903)/ UAS-Chk1^{S373D})* larvae at 16–24 hr L3 (n ≥ 7 tracheae per condition per timepoint). Student's t-test: *p<0.00001.

The online version of this article includes the following figure supplement(s) for figure 3:

**Source data 1.** Cell frequencies in Chk1S373D expressing animals.

no obvious slowdown in cell division rate after mitotic re-entry (*Figure 2G and H*, n ≥ 7 tracheae per timepoint, *Figure 2—source data 1*, *Figure 2—source data 2*).

## ROS dependence reveals a novel mode of ATR/Chk1 regulation

Having identified the source for high ROS in arrested tracheoblasts, we turned our attention to addressing how ROS is coordinating G2 arrest. Our previous studies have shown that Wnt-dependent transcriptional upregulation of Chk1 is essential for G2 arrest. Wnt signaling in the trachea is mediated by four Wnt ligands – Wg, Wnt5, Wnt6, and Wnt10 – that are expressed by the tracheoblasts (*Figure 3A*). All ligands are expressed at high levels in arrested cells and downregulated post-mitotic entry. We have also shown that the four Wnts act synergistically to upregulate Chk1 expression but are redundant for expression of other Wnt targets like Fz3.

Our first step toward characterizing the role of ROS in G2 arrest was to analyze how ROS levels impacted Wnt signaling and the expression of Wnt target genes, particularly Chk1. We micro-dissected Tr2 fragments from *btl-SOD1* and *btl-Duox^{RNAi}* animals at L2, extracted mRNA, and analyzed expression of Wnts and Wnt target genes by qPCR. We found that the expression of all Wnt ligands, Fz3 and Chk1, was comparable in wild type, *btl-SOD1* and *btl-Duox^{RNAi}*-expressing animals (*Figure 3B and C*, red dashed line marks wild type levels, n ≥ 15 tracheal fragments per timepoint per experiment, n = 3 experiments). This showed that perturbations in ROS levels in the trachea do not impact Wnt signaling nor expression of Wnt targets like Chk1. We also assayed the levels of ATR in *btl-SOD1* and *btl-Duox^{RNAi}*-expressing animals by qPCR and found no change in ATR transcript levels compared to control (*Figure 3B and C*, red dashed line marks wild type levels). Together, the qPCR data suggested that ROS did not regulate the abundance of either Chk1 or ATR transcripts.

We have shown previously that precocious mitotic re-entry observed in Wnt signaling-deficient tracheoblasts (*btl-TCF^{RNAi}*) can be rescued by overexpression of Chk1 (*Kizhedathu et al., 2020*). Thus, to functionally test whether ROS levels impact Chk1 expression, we overexpressed Chk1 in *btl-Duox^{RNAi}*-expressing animals and examined cell proliferation. We counted numbers of tracheoblasts in *btl-GAL4/+; UAS-Duox^{RNAi}/UAS-Chk1* (*btl-Duox^{RNAi}*, *Chk1*) animals at 16–24 hr L3 to find that the numbers were considerably higher than wild type and comparable to the numbers in *btl-Duox^{RNAi}* animals (*Figure 3D*, n ≥ 7 tracheae, *Figure 3—source data 1*). Along these lines, we also overexpressed ATR in *btl-Duox^{RNAi}* animals and counted the number of Tr2 tracheoblasts at 16–24 hr L3. We found that the numbers were higher than wild type and comparable to the numbers in *btl-Duox^{RNAi}* animals (*Figure 3D*, n ≥ 7 tracheae, *Figure 3—source data 1*). Taken together, the analyses suggested that ROS does not regulate ATR/Chk1 gene expression.

In light of the findings that ROS depletion does not perturb Chk1 expression but does perturb Chk1 phosphorylation and function, we hypothesized that ROS may regulate Chk1 phosphorylation in some manner. The ATR-dependent phosphorylation of Chk1 at serine 373 is thought to be necessary for its activation (*Liu et al., 2000*; *Patil et al., 2013*; *Bayer et al., 2018*). Our immunohistochemical analyses are consistent with these findings. To probe the possibility that ROS facilitates Chk1 phosphorylation, we tested whether a phosphomimic variant of Chk1, in which the serine at the position 373 has been replaced by aspartic acid (Chk1^{S373D}), could rescue the Duox phenotype. We overexpressed Chk1^{S373D} in *btl-Duox^{RNAi}* animals and counted cell numbers in Tr2 at 16–24 hr L3. We found that Tr2 cell numbers in these animals were now comparable to wild type (and lower than in *btl-Duox^{RNAi}* animals, *Figure 3D*, n ≥ 7 tracheae, *Figure 3—source data 1*). The rescue of the Duox mutant phenotype by the phosphomimic variant of Chk1 suggested that ROS is required for ATR-dependent phosphorylation and activation of Chk1.

## ROS-dependent activation of ATR/Chk1 does not require ATRIP/TOPBP1/claspin

The coincidence of high ROS levels and activated Chk1 in cells would typically implicate ROS-dependent genotoxic stress as the driver of Chk1 activation. However, our analysis of DNA damage in tracheoblasts, using the well-characterized marker for double-strand DNA breaks (γ-H2AX), found no detectable DNA damage in arrested cells (*Kizhedathu et al., 2018*). In light of the findings with respect to ROS, we decided to probe more rigorously the incidence of genotoxic stress in tracheoblasts and the role of the DNA damage response in Chk1 activation.

We re-evaluated the levels of DNA damage in tracheoblasts using two assays. First, we examined the accumulation of 8-oxo-2'-deoxyguanosine (8-oxo-dG), a marker for nucleotide oxidation. Second, we scored the frequencies of nuclear foci of RPA70, a protein that binds single-strand DNA breaks. To validate nuclear 8-oxodG as a marker of oxidative damage in tracheoblasts, tracheae from L2 larvae were dissected and treated ex vivo with different concentrations of $H_2O_2$ (100 µM, 500 µM, and 1 mM) for 30 min and stained with an antibody against 8-oxodG. Robust staining was detected in trachea at 1 mM $H_2O_2$(***Figure 4A***, n ≥ 6 tracheae per condition per experiment, n = 2), but no signal was detected at lower concentrations or in untreated tracheae. This showed that although 8-oxodG accumulation is responsive to oxidative stress, there is no 8-oxodG accumulation in G2-arrested tracheoblasts under normal conditions. An aspect of the 8-oxodG staining in tracheal cells merits mention here. The accumulation of 8-oxodG in Tr2 tracheoblasts was cytoplasmic unlike the tracheal cells in other metameres, where 8-oxodG was observed in both cytoplasm and nucleus (***Figure 4B***, n ≥ 6 tracheae per condition per experiment, n = 2). One reason for this difference could be that Tr2 DT are arrested in G2 and not engaged in DNA synthesis while cells in other metameres are actively endocycling and replicating DNA.

Next we probed the incidence of single-stranded DNA breaks in tracheoblasts with the help of a strain that ubiquitously expresses RPA70-GFP (***Blythe and Wieschaus, 2015***). RPA 70 has been shown to be uniformly distributed in the nucleus under normal conditions and to form focal nuclear aggregates at sites of single-strand DNA breaks (***Blythe and Wieschaus, 2015***). To validate RPA*70-GFP* as a marker for genotoxic stress in the tracheal system, L2 animals were exposed to either 0 (control) or 50 Gy of γ-irradiation and immunostained for GFP. Foci of GFP could be observed in the nuclei of tracheoblasts exposed to 50 Gy γ-irradiation (***Figure 4C***, n ≥ 6 tracheae per condition per experiment, n = 3) but no foci were detected in untreated tracheae at the same stage. In a parallel set of experiments, we also examined levels of γ-H2AX in L2 animals under these conditions. We observed foci of nuclear γ-H2AX staining in tracheoblasts exposed to 50 Gy of γ-irradiation (***Figure 4D***, n ≥ 6 tracheae per condition per experiment, n = 3) but not in untreated tracheae at the same stage. Taken together, our analysis of 8-oxodG, RPA70-GFP, and γ-H2AX further confirmed that there is no detectable DNA damage in arrested tracheoblasts.

To probe the relationship between DNA damage and Chk1 activation in tracheoblasts, we also utilized a genetic approach. The mechanism for ATR/Chk1 activation in response to DNA damage has been characterized in some detail. These studies show that the activation of ATR/Chk1 requires three major proteins: ATR interacting protein (ATRIP, *mus-304*), topoisomerase II binding protein 1 (TOPBP1, *mus-101*), and claspin. Breaks in DNA that are bound by single-strand DNA-binding proteins like RPA-1/RPA-70 recruit ATR via its partner ATRIP and, in turn, TOPBP1. This complex activates ATR, and consequently, in a claspin-dependent manner, Chk1 (***Choi et al., 2010***; ***Cimprich and Cortez, 2008***; ***Delacroix et al., 2007***; ***Lee et al., 2012***; ***Xu and Leffak, 2010***).

To determine if ATRIP, TOPBP1, and claspin are required in tracheal cells for DNA damage-dependent phosphorylation of Chk1, we examined pChk1 levels post γ-irradiation in tracheoblasts in which we simultaneously knocked down Duox and the aforementioned gene products. We exposed animals expressing *btl-Duox^RNAi^*, *ATRIP^RNAi^* (btl-GAL4/ UAS-ATRIP^RNAi^; UAS-Duox^RNAi^/+), *btl-Duox^RNAi^*, *TopBP1^RNAi^* (btl-GAL4/+; UAS-Duox^RNAi^/UAS-TOPBP1^RNAi^), and *btl-Duox^RNAi^*, *btl-Duox^RNAi^*, *Claspin^RNAi^* (btl-GAL4/+; UAS-Duox^RNAi^/UAS-Claspin^RNAi^) to 50 Gy of γ-radiation and performed immunostaining for pChk1 1 hr after irradiation (***Figure 4—figure supplement 1A***). Although pChk1 could be detected post irradiation in animals expressing *btl-Duox^RNAi^* (***Figure 5—figure supplement 1C***, n ≥ 6 tracheae per condition per experiment, n = 3), we did not detect pChk1 in *btl-Duox^RNAi^*, *ATRIP^RNAi^*, *btl-Duox^RNAi^*, *TOPBP1^RNAi^*, and *btl-Duox^RNAi^*, *Claspin^RNAi^*-expressing animals at the same stages (***Figure 4—figure supplement 1B–E***, n ≥ 6 tracheae per condition per experiment, n = 2). This shows that DNA damage-dependent phosphorylation of Chk1 in Tr2 DT cells requires ATRIP, TOPBP1, and claspin.

To determine if any of the components of DNA damage-dependent ATR activation are necessary for the phosphorylation of Chk1 in the trachea, we knocked down ATR, ATRIP, TOPBP1, and claspin and probed the levels of pChk1 in L2 (***Figure 4E–H***, n ≥ 6 tracheae per condition per experiment, n = 2). pChk1 staining of the tracheae from these animals revealed that the loss of ATR led to the loss of pChk1 (***Figure 4E***). In contrast, the knockdown of ATRIP, TOPBP1, or claspin did not lead to a loss of pChk1 in L2 (***Figure 4F–H***). We also counted the number of cells in Tr2 DT at L2 and 16–24 hr L3 in each of these genetic backgrounds. While the knockdown of ATR led to an increase in cell number

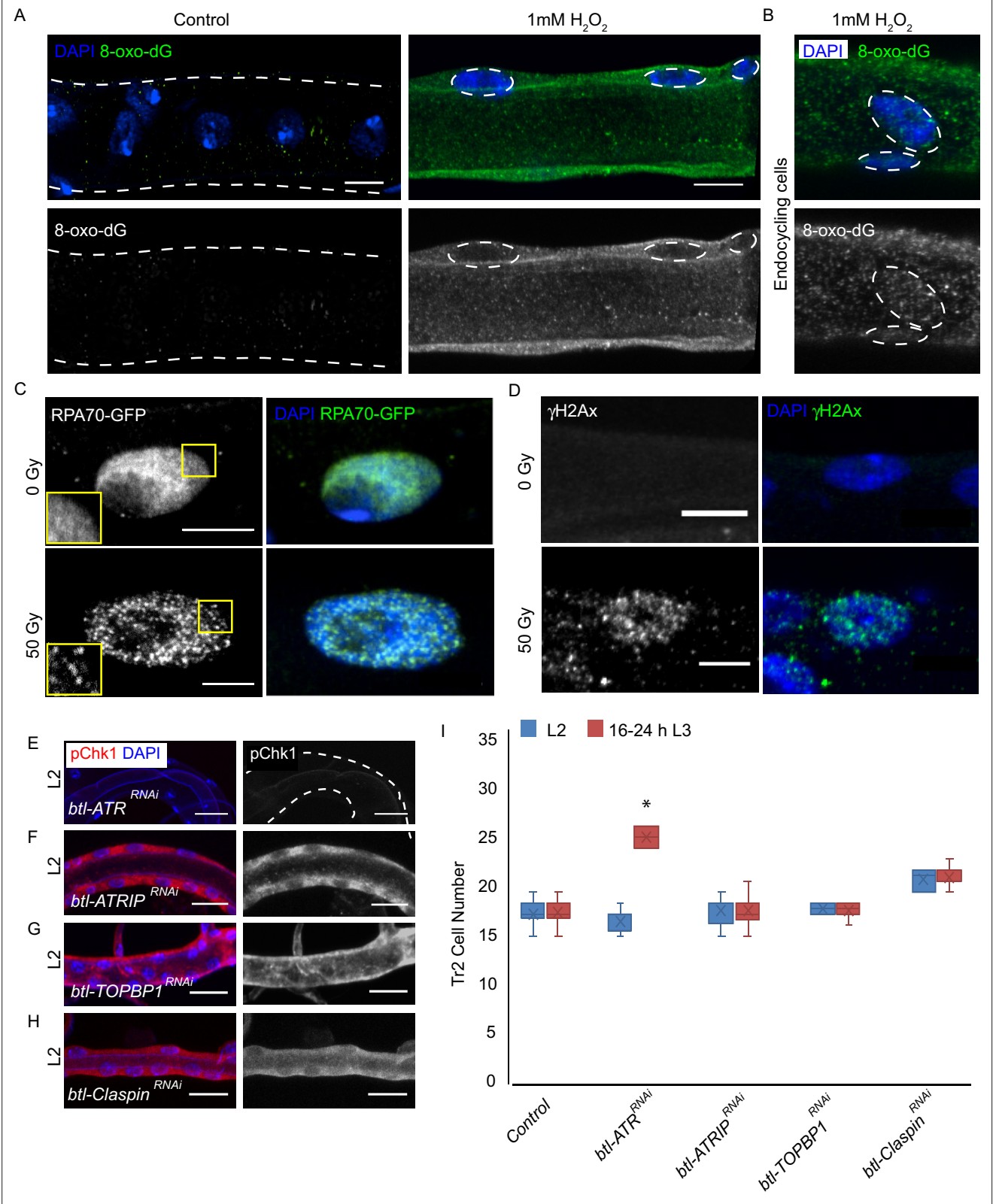

**Figure 4.** ATRIP, TOPBP1, and claspin are not required for reactive oxygen species (ROS)-mediated checkpoint kinase 1 (Chk1) activation in tracheoblasts. (**A–D**) Detailed analysis of DNA damage in Tr2 dorsal trunk (DT). Shown here are findings from three different reporters of genotoxic stress. (**A**) 8-Oxo-2'-deoxyguanosine (8-Oxo-dG) immunostaining in wild type (*btl-GAL4*) Tr2 DT in untreated tracheae (left panel) and tracheae exposed to 1 mM H$_2$O$_2$ for 30 min ex vivo (right panel) at L2. (**B**) 8-Oxo-dG immunostaining in wild type (*btl-GAL4*) endocycling cells of the tracheae exposed

*Figure 4 continued on next page*

*Figure 4 continued*

to 1 mM $H_2O_2$ for 30 min ex vivo at L2. (**C**) GFP immunostaining in non-irradiated and $\gamma$-irradiated larvae expressing RPA70-GFP. Shown in the figure are GFP immunostaining in non-irradiated larvae (top panel) and larvae exposed to 50 Gy of $\gamma$-radiation (bottom panel) at L2. (**D**) $\gamma$-H2AX$^{Ser139}$ immunostaining in Tr2 DT in wild type (*btl-GAL4*) non-irradiated larvae (top panel) and larvae irradiated with 50 Gy of $\gamma$-radiation (bottom panel) at L2. (**E–H**) Analysis of the contribution of components of the DNA damage-dependent activation of ATR/Chk1 to Chk1 activation in Tr2 DT. Effects of the knockdown of *ATR, ATRIP, TOPBP1,* and *Claspin* on phosphorylated checkpoint kinase 1 (pChk1) levels in Tr2 DT at L2. pChk1 immunostaining (red) in Tr2 DT in (**E**) *btl-ATR$^{RNAi}$* (*btl-GAL4/UAS-ATR$^{RNAi}$*), (**F**) *btl-ATRIP$^{RNAi}$* (*btl-GAL4/UAS-ATRIP$^{RNAi}$*), (**G**) *btl-TOPBP1$^{RNAi}$* (*btl-GAL4/+; UAS-TOPBP1$^{RNAi}$/+*), and (**H**) *btl-Claspin$^{RNAi}$* (*btl-GAL4/+; UAS-Claspin$^{RNAi}$/+*) larvae at L2. (**I**) Effects of knockdown of *ATR, ATRIP, TOPBP1,* and *Claspin* on cell numbers in Tr2 DT. Graph shows numbers of Tr2 tracheoblasts in wild type (*btl-Gal4*), *btl-ATR$^{RNAi}$* (*btl-GAL4/UAS-ATR$^{RNAi}$*), *btl-ATRIP$^{RNAi}$* (*btl-GAL4/UAS-ATRIP$^{RNAi}$*), *btl-TOPBP1$^{RNAi}$* (*btl-GAL4/+; UAS-TOPBP1$^{RNAi}$/+*), and *btl-Claspin$^{RNAi}$* (*btl-GAL4/+; UAS-Claspin$^{RNAi}$/+*) at L2 and 16–24 hr L3 (mean values ± standard deviation, n ≥ 7 tracheae per condition per timepoint. Scale bars = 5 μm (**A–D**), 10 μm (**E–H**). Student's t-test: *p<0.00001.

The online version of this article includes the following figure supplement(s) for figure 4:

**Source data 1.** Cell frequencies in ATR $^{RNAi}$, ATRIP $^{RNAi}$, TOPBP1 $^{RNAi}$ and Claspin $^{RNAi}$ expressing animals.

**Figure supplement 1.** ATRIP, TOPBP1, and claspin are required for DNA damage-dependent activation ofATR/Chk1.

**Figure supplement 2.** Loss of ATRIP, TOPBP1, and claspin does not affect cell numbers at WL3.

at 16–24 hr L3, knockdown of ATRIP, TOPBP1, and claspin did not (***Figure 4I***, n ≥ seven tracheae per timepoint, ***Figure 4—source data 1***). These data indicate that ATR-dependent activation of Chk1 in G2-arrested tracheoblasts cells does not require ATRIP, TOPBP1, or claspin.

Collectively, these experiments lend support to the idea that the ROS-dependent Chk1 activation in tracheoblasts does not involve the DNA damage response pathway.

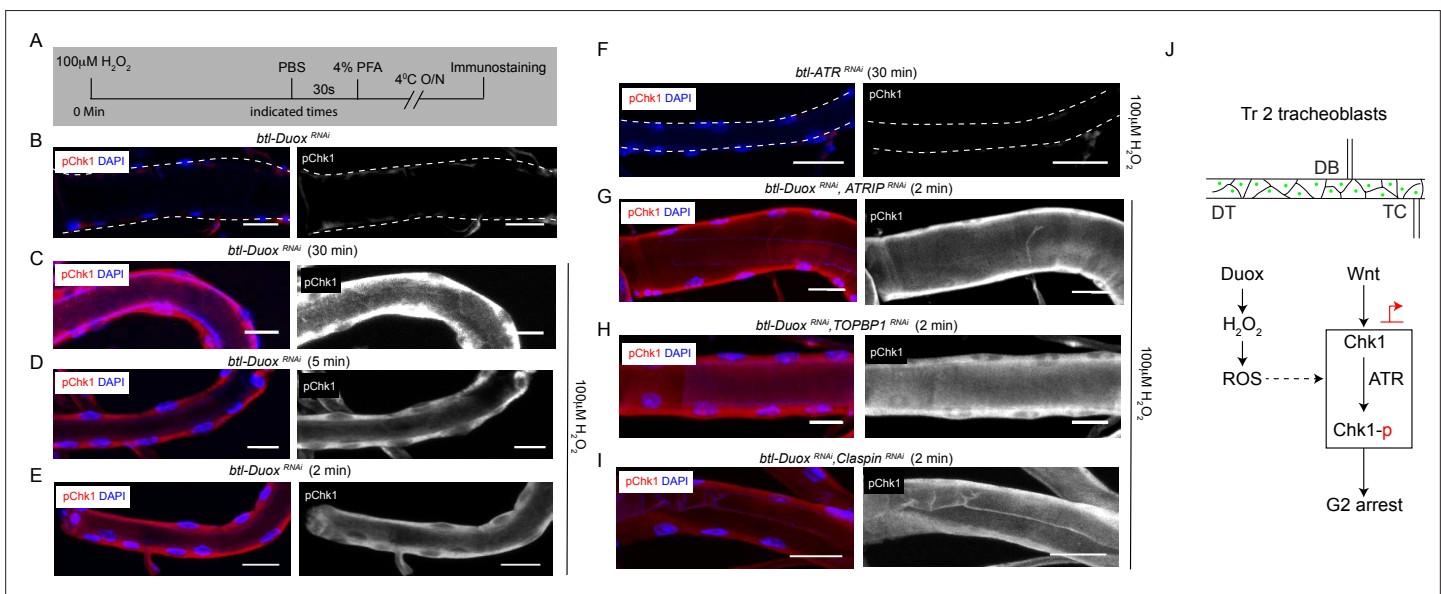

**Figure 5.** Incubation with $H_2O_2$ can restore phosphorylated checkpoint kinase 1 (pChk1) levels in Dual oxidase (Duox)-deficient tracheoblasts. (**A–E**) Kinetics of Chk1 phosphorylation upon exposure to $H_2O_2$ ex vivo. (**A**) Regimen for $H_2O_2$ treatment and analysis of pChk1 in Tr2 dorsal trunk (DT) in L2. pChk1 immunostaining (red) in Tr2 DT in (**B**) untreated *btl-Duox$^{RNAi}$* (*btl-GAL4/+; UAS-Duox$^{RNAi}$ (32903)/+*)-expressing tracheae and treated with 100 μM $H_2O_2$ for (**C**) 30 min, (**D**) 5 min, and (**E**) 2 min. (**F**) Effect of knockdown of ATR on Chk1 activation in Tr2 DT upon exposure to $H_2O_2$ ex vivo. pChk1 immunostaining (red) in Tr2 DT in *btl-ATR$^{RNAi}$* (*btl-GAL4/UAS-ATR$^{RNAi}$*) tracheae treated with 100 μM $H_2O_2$ for 30 min. (**G–I**) Effect of knockdown of *Duox* and *ATRIP* or *TOPBP1* or *Claspin* on pChk1 levels in Tr2 DT in tracheae exposed to 100 μM $H_2O_2$ at L2. pChk1 immunostaining (red) in Tr2 DT in (**G**) *btl-Duox$^{RNAi}$, ATRIP$^{RNAi}$* (*btl-GAL4/ UAS-ATRIP$^{RNAi}$; UAS-Duox$^{RNAi}$ (32903)/+*), (**H**) *btl-Duox$^{RNAi}$, TOPBP1$^{RNAi}$* (*btl-GAL4/+; UAS-Duox$^{RNAi}$(32903)/UAS-TOPBP1$^{RNAi}$*) and (**I**) *btl-Duox$^{RNAi}$, Claspin$^{RNAi}$* (*btl-GAL4/+; UAS-Duox$^{RNAi}$(32903)/UAS-Claspin$^{RNAi}$*) tracheae treated with 100 μM $H_2O_2$ for 2 min at L2. (**J**) Model for the regulation of ATR/Chk1 activation in Tr2 DT. We propose that $H_2O_2$ can induce ATR-dependent phosphorylation and activation of Chk1 in the absence of detectable DNA damage, leading to G2 arrest in Tr2 tracheoblasts. Scale bars = 10 μm.

The online version of this article includes the following figure supplement(s) for figure 5:

**Figure supplement 1.** Exposure to $\gamma$-radiation can restore phosphorylated checkpoint kinase 1 (pChk1) levels in dual oxidase (Duox)-deficient tracheoblasts.

## H$_2$O$_2$ can rescue pChk1 levels in Duox-deficient tracheoblasts

The next obvious question was to ask if Chk1 phosphorylation can be induced in Duox mutants by the addition of H$_2$O$_2$. To investigate this possibility, we examined levels of pChk1 in *btl-Duox$^{RNAi}$* tracheae after exposure to different concentrations of H$_2$O$_2$ for different periods of time. Tracheae from L2 animals were exposed to PBS or H$_2$O$_2$ (PBS) ex vivo and immunostained for pChk1 (*Figure 5A*). We detected no pChk1 staining in tracheae exposed to buffer alone and robust pChk1 staining in tracheae incubated with H$_2$O$_2$ (*Figure 5B–E*, *Figure 1—figure supplement 2C*, n ≥ 6 tracheae per condition per experiment, n = 3). Interestingly, we noted that exposure to H$_2$O$_2$ for periods as short as 2 min was sufficient to restore levels of pChk1 in *btl-Duox$^{RNAi}$* tracheae (*Figure 5E*). We also probed pChk1 levels in *btl-ATR$^{RNAi}$ (btl-GAL4/UAS-ATR$^{RNAi}$)* tracheae post H$_2$O$_2$ treatment and found that there was no pChk1 accumulation (*Figure 5G*, *Figure 1—figure supplement 2C*, n ≥ 6 tracheae per condition per experiment, n = 3). Together, these data show that H$_2$O$_2$ treatment is sufficient to induce Chk1 phosphorylation in an ATR-dependent manner and that H$_2$O$_2$ can induce pChk1 in minutes.

In an independent set of experiments, we examined the kinetics of DNA damage-dependent activation of Chk1 in tracheoblasts. As described earlier, we exposed L2 larvae to 50 Gy of γ-radiation and performed pChk1 immunostaining at different timepoints post irradiation (*Figure 5—figure supplement 1A*). We could detect pChk1 1 hr post irradiation (*Figure 5—figure supplement 1C*, n ≥ 6 tracheae per condition per experiment, n = 2) but not earlier (*Figure 5—figure supplement 1D* and E, n ≥ 6 tracheae per condition per experiment, n = 2). Here again, pChk1 induction in response to γ-radiation was dependent on ATR as pChk1 was not detected in tracheoblasts expressing *btl-ATR$^{RNAi}$* (*Figure 5—figure supplement 1F*, n ≥ 6 tracheae per condition per experiment, n = 2). These experiments suggest that the kinetics of Chk1 phosphorylation in response to H$_2$O$_2$ could be significantly faster than in response to γ-radiation.

Next, we tested whether H$_2$O$_2$ could restore levels of pChk1 in the absence of ATRIP, TOPBP1, and claspin in *btl-Duox$^{RNAi}$*-expressing animals. To test this possibility, we exposed animals expressing *btl-Duox$^{RNAi}$, ATRIP$^{RNAi}$, btl-Duox$^{RNAi}$, TOPBP1$^{RNAi}$*, and *btl-Duox$^{RNAi}$, Claspin$^{RNAi}$*-expressing tracheae to 100 μM H$_2$O$_2$ and performed Chk1 immunostaining 2 min after exposure. pChk1 immunostaining showed that the knockdown of ATRIP, TOPBP1, and claspin did not prevent the activation of Chk1 (*Figure 5G–I*, *Figure 1—figure supplement 2C*, n ≥ 6 tracheae per condition per experiment, n = 2). This strongly suggests that the H$_2$O$_2$-dependent phosphorylation of Chk1 is independent of ATRIP, TOPBP1, and claspin.

## Discussion

ATR and Chk1 are essential for normal development in *Drosophila* and other animals. In this regard, both kinases are thought to serve as guardians of genomic integrity and loss of either is associated with increased genomic instability and catastrophic cell death. Our studies in the tracheal system reveal a different facet of ATR/Chk1 function . We have shown previously that the pathway is required to arrest tracheal progenitor cells in G2 and that the activation of ATR/Chk1 occurs in the absence of any detectable DNA damage. The findings presented here demonstrate that Duox-generated H$_2$O$_2$ is required for the activation of ATR/Chk1 axis in this context (see model in *Figure 5J*). We discuss below the possible mechanisms by which ROS can activate ATR/Chk1, how ROS levels are regulated during tracheal development, and the clinical implication of the findings.

The precedence for ROS-based activation of PIKK-family kinases like ATR, ATM, and DNA-PK was set by studies on ATM. The formation of an intermolecular disulfide bond between cysteine residues located at the C-terminus of ATM was found to be essential for ATM homodimerization and activation (*Guo et al., 2010*). The structure of *Drosophila* ATR is not known, but the structure of human ATR in complex with ATRIP has been determined by cryoEM and the human ATR-ATRIP complex has been shown to dimerize (*Rao et al., 2017*). Using SWISS-MODEL and the kinase domain of human ATR as template, we modeled the *Drosophila* ATR monomer (the sequence range modeled was 855–2517, which has 33.6% sequence identity to human ATR). A model for the *Drosophila* ATR dimer was subsequently generated using the human ATR-ATRIP complex as the template (*Waterhouse et al., 2018*; *Guex et al., 2009*). This model shows that there are no exposed cysteines that are close enough to form intermolecular disulfide bridges. Thus, there is a possibility that H$_2$O$_2$ regulates ATR in a different way than what has been proposed for ATM. The possibility that H$_2$O$_2$ can directly activate ATR by

other mechanisms seems likely. ROS can alter kinase activity either by modifying cysteine residues at or near the active site or more broadly, leading to conformational changes (*Corcoran and Cotter, 2013*). As indicated in the Introduction, ATR can be activated in response to mechanical stress by a mechanism that is likely to be dependent on ATRIP. Although our studies show that activation of ATR/Chk1 by ROS does not require ATRIP, future experiments will investigate all possibilities for non-canonical activation. ATR aside, ROS may also regulate Chk1 activation by modifying Chk1 in some manner or via recruitment of other factors (see model in *Figure 5J*).

A comparison of the effects of ATR/Chk1 knockdown and SOD1 overexpression/Duox knockdown suggests that there may be other (ROS-independent) mechanisms for the non-canonical activation of ATR/Chk1. We have shown that loss of either ATR or Chk1 leads to slow rate of cell division post mitotic reentry (*Kizhedathu et al., 2018*). Since there is no evidence for any DNA damage post-mitotic re-entry and the knockdown of ATRIP, TOPBP1, and claspin does not recapitulate the ATR/Chk1 mutant phenotype (*Figure 4—figure supplement 2*, n ≥ 7 tracheae per timepoint), the mechanism for the activation of ATR/Chk1 post-mitotic re-entry is unclear. Interestingly, neither the overexpression of SOD1 nor the knockdown of Duox recapitulates the mitotic defect observed in ATR/Chk1 mutants (*Figures 1J and 2F*). This suggests that there are other non-canonical-mechanisms of ATR/Chk1 activation that are relevant to the roles of these kinases during development.

How are ROS levels regulated during development? Analysis of Duox mRNA levels in tracheoblasts shows that expression is high in L2 and early L3 and drops significantly at 32–40 hr L3. This parallels the expression of Chk1. We have previously shown that the levels of Chk1 mRNA are regulated transcriptionally by the Wnt signaling pathway. We determined whether Wnt signaling also regulates Duox expression and ROS levels. This analysis showed that ROS levels are unaffected in Wnt pathway mutants (data not shown). We infer that Wnt signaling pathway does not regulate Duox nor ROS levels. Juvenile hormone (JH) has been shown to act as a negative regulator of cell proliferation in tracheae (*Djabrayan et al., 2016*). Pertinently, the levels of JH are high in L2 and early L3 and drop mid L3 (*Dubrovsky, 2005*). Thus, the timecourse of JH levels parallels the timecourses of both Duox and Chk1 expression. Future experiments will test the possibility that JH signaling in trachea regulates Duox expression and potentially Wnt signaling/Chk1 expression as well.

The possibility that ROS can activate ATR/Chk1 without inducing DNA damage may be clinically relevant. It has been reported that ovarian cancer cells that express high levels of Duox1 and high levels of activated Chk1 are relatively Cisplatin-resistant (*Meng et al., 2018*). The authors suggest that high levels of Chk1 activation are critical for chemoresistance and that the high levels of Chk1 activation are likely the result of ROS-generated DNA damage. The findings presented here suggest that high ROS may independently activate Chk1 and contribute toward chemoresistance. Along the same lines, cancer cells that have high levels of ROS (*Gu et al., 2018*; *Singh et al., 2020*) and an active DNA damage repair pathway have been shown to be relatively radioresistant (*Alsubhi et al., 2016*; *Wang et al., 2013*). Here again, the enhanced radioresistance has been attributed to ROS-dependent DNA damage priming of the DNA damage response. We suggest that the ROS-based activation of ATR/Chk1 activation evidenced here may also contribute toward the observed chemo-/radioresistance of cancer cells.

# Materials and methods

## Key resources table

| Reagent type (species) or resource | Designation | Source or reference | Identifiers | Additional information |
|---|---|---|---|---|
| Genetic reagent (*Drosophila melanogaster*) | btl-GAL4 | *Shiga et al., 1996* | FLYB: FBtp0001208 | This line was a gift from Dr. Shigeo Hayashi |
| Genetic reagent (*D. melanogaster*) | UAS-Chk1$^{RNAi}$ | VDRC | 110076 | |
| Genetic reagent (*D. melanogaster*) | UAS-Chk1$^{S373D}$ | This study | | Please see Materials and methods for a detailed description. (Can be obtained through NCBS Fly Facility: https://bangalorefly.ncbs.res.in/) |

*Continued*

| Reagent type (species) or resource | Designation | Source or reference | Identifiers | Additional information |
|---|---|---|---|---|
| Genetic reagent (*D. melanogaster*) | *RPA-70GFP* | **Blythe and Wieschaus, 2015** | | This line was a gift from Dr. Eric F Wieschaus |
| Genetic reagent (*D. melanogaster*) | *UAS-Duox*^RNAi | BDSC | RRID:BDSC_33975 and RRID:BDSC_32903 | |
| Genetic reagent (*D. melanogaster*) | *UAS-ATR* | **Bayer et al., 2018** | | This line was a gift from Dr. Anja C Nagel |
| Antibody | Phospho-Chk1 (Ser345) (rabbit monoclonal) antibody | CST | Cat #2348 (RRID:AB_331212) | (1:200) |
| Antibody | Anti-8-hydroxy-2'-deoxyguanosine antibody (mouse monoclonal) antibody | Abcam | Cat #ab48508 (RRID:AB_867461) | (1:200) |
| Commercial assay or kit | Tyramide signal amplification system | Thermo Fisher Scientific | Cat #B40912 | |

## Fly strains and handling

The following strains were obtained from repositories: *UAS-Sod1* (RRID:BDSC_24754), UAS-*Duox*^RNAi (RRID:BDSC_32903, RRID:BDSC_33975), UAS-*ATRIP*^RNAi (RRID:BDSC_61355), *UAS-TOPBP1*^RNAi (RRID:BDSC_43244), UAS-*Claspin*^RNAi (RRID:BDSC_32974) (Bloomington *Drosophila* Stock Center), *UAS-Chk1*^RNAi (RRID:FlyBase_FBst0473748), and *UAS-ATR*^RNAi (RRID:FlyBase_FBst0475838; Vienna *Drosophila* Resource Center). *UAS-Chk1* was generated in the in-house fly facility. The following strains were received as gifts: *btl-GAL4, UAS-ATR*, and *RPA70-GFP*. Strains were raised on a diet of cornmeal-agar and maintained at 25°C . All experiments were performed on animals raised at 25°C unless otherwise indicated.

## Cloning of pUAST-Chk1^S373D and generation of transgenic flies

*Drosophila* Chk1 cDNA clone was ordered from DGRC in pOT2 vector. Serine at 373 (TCG) position was mutated to aspartic acid (GAT) using the primers *Chk1_S373D_Forward* and *Chk1_S373D_Reverse*. The complete plasmid was amplified by polymerase chain reaction (PCR) using Phusion polymerase. The PCR product was then digested with Dpn1 enzyme and further transformed into XL10 cells. The plasmid was isolated from a few colonies and sent for sequencing. The positive clone with the mutation (TCG to GAT) was then subcloned into the vector pUAST. Mutant Chk1 (Chk1 S373D) was amplified using the primers *pUAST_Chk1_Ecor1_Forward* and *pUAST_Chk1_Kpn1_Reverse*. The PCR fragment and the empty vector were then double digested using enzymes EcoR1 and Kpn1 at 37°C for 1 hr. The vector was purified using gel extraction and the PCR fragment was purified using the PCR clean-up kit (Qiagen). The digested vector and insert were mixed in the ratio 1:3 and ligated using T4 DNA ligase (NEB) at 16°C overnight. The ligation mixture was transformed into XL10 cells. Plasmid isolation was performed on the positive clones and sent for sequencing for further confirmation. The clone with the correct mutation pUAST-Chk1^S373D was used to establish five independent transgenic fly lines by P-element mediated germline transformation by the in-house fly facility.

The following primer sets were used for generating and cloning Chk1^S373D:

| | |
|---|---|
| *Chk1_S373D_Forward* | 5' CAGTTACTCCTTCGATCAACCAGCTTTGCTTGATG 3' |
| *Chk1_S373D_Reverse* | 5' ATCGAAGGAGTAACTGAGCCGAGCCTCCTG 3' |
| *pUAST_Chk1_EcoR1_Forward* | 5' AGAGAATTCATGGCTGCAACGCTG 3' |
| *pUAST_Chk1_Kpn1_Reverse* | 5' AGAGGTACCCTAAGGCACCGAATTTG 3' |

## Larval staging

Larval staging was performed as previously described (*Guha and Kornberg, 2005*) based on the morphology of the anterior spiracles. L2 larvae were collected and examined to identify animals that had undergone the L2-L3 molt in 8 hr intervals (0–8 hr L3). 0–8 hr L3 cohorts collected in this method were staged for subsequent timepoints.

## Immunostaining and imaging

Animals were dissected in PBS and fixed for 30 min with 4% (w/v) paraformaldehyde (PFA) in PBS. The following antisera were used for immunohistochemical analysis: chicken anti-GFP (Aves, 1:500, RRID:AB_10000240), rabbit anti-phospho Chk1 (CST, 1:200, RRID:AB_331212), rabbit anti-pH3 (Millipore, 1:500, RRID:AB_310177), mouse anti-8-hydroxy-2'-deoxyguanosine (Abcam, 1:200, RRID:AB_867461), and Alexa 488/568-conjugated donkey anti-chicken/rabbit/mouse secondary antibodies (Invitrogen, 1:200). Tyramide signal amplification was performed as per the manufacturer's recommendations for pChk1 detection. The following reagents were used as part of this protocol: tyramide amplification buffer and tyramide reagent (Thermo Fisher), vectastain A and B (Vector Labs), and biotinylated donkey anti-rabbit IgG (Jackson ImmunoResearch, 1:200, RRID:AB_2340593). Tracheal preparations were flat-mounted in ProLong Diamond Antifade Mountant with DAPI (Molecular Probes) and imaged on Zeiss LSM-780 laser-scanning confocal microscopes. All images were taken by adjusting the parameters (gain and laser power) such that there is no saturation in the positive control images. All images from the same experiment were acquired at the same settings. Images were processed using ImageJ (RRID:SCR_003070). For quantification of cell number, fixed specimens were mounted in ProLong Diamond Antifade Mountant with DAPI and the number of nuclei was counted from images collected with an Olympus BX 53 microscope. The DT of the second thoracic metamere was identified morphologically based on the cuticular banding pattern at anterior and posterior junctions.

## ROS detection

Larvae of indicated stages were dissected in PBS, flipped inside out to expose the trachea, and incubated in 100 μM $H_2$DCFDA (Thermo Fisher) for 30 min or 10 μM DHE (Thermo Fisher) for 5 min at room temperature. The larvae were then washed in PBS and fixed mildly in 4% PFA for 5 min. Tracheae were flat mounted in ProLong Diamond and imaged immediately.

## Fluorescence intensity quantification

Fluorescence intensities were quantified by doing a maximum intensity projection followed by background subtraction. In each image, three square regions of interest (ROIs) were selected at random within the tracheal boundaries. The intensity density values obtained from these ROIs were averaged and then divided by 1000 to obtain arbitrary unit values (AU). Fluorescence intensities of all the samples for $H_2$DCFDA, DHE staining, and pChk1 immunostaining were calculated in the same manner. ImageJ software was used to perform all the above operations.

## RNA isolation and quantitative PCR

RNA extraction and qPCR were performed as described in *Kizhedathu et al., 2018*. Primer sequences for *Chk1*, *Fz3*, *Wg*, *Wnt5*, *Wnt6*, *Wnt10*, *ATR*, *Duox*, and *GAPDH* (internal control) are provided below. Relative mRNA levels were quantified using the formula RE = $2^{-\Delta\Delta Ct}$ method.

The following primer sets were used:

| | |
|---|---|
| *GAPDH* forward | 5' CGTTCATGCCACCACCGCTA 3' |
| *GAPDH* reverse | 5' CACGTCCATCACGCCACAA 3' |
| *Chk1* forward | 5' AACAACAGTAAAACGCGCTGG 3' |
| *Chk1* reverse | 5' TGCATATCTTTCGGCAGCTC 3' |
| *Wg* forward | 5' AAATCGTTGATCGAGGCTGC 3' |
| *Wg* reverse | 5' GGTGCAGGACTCTATCGTTCC 3' |

*Continued on next page*

*Continued*

| | |
|---|---|
| *Wnt5* forward | 5' AGGATAACGTGCAAGTGCCA 3' |
| *Wnt5* Reverse | 5' ACTTCTCGCGCAGATAGTCG 3' |
| *Wnt6* Forward | 5' AGTTTCAATTCCGCAACCGC 3' |
| *Wnt6* Reverse | 5' TCGGGAATCGCGCATTAAGA 3' |
| *Wnt10* Forward | 5' CACGAATGGCCCGAAAACTG 3' |
| *Wnt10* Reverse | 5' CCCACGGTGCCCTGTATATC 3' |
| *Fz3* Forward | 5' ATGAATGTCGTTCAAAGTGG 3' |
| *Fz3* Reverse | 5' TATAGTAAATGGGGCTTGCG 3' |
| *ATR* Forward | 5' CCAGATAGCAGCGAGTGCAT 3' |
| *ATR* Reverse | 5' CGAGGTCCAGGGAACTTAGC 3' |
| *Duox* Forward | 5' ATCTACACGGTGGATAGGAA 3' |
| *Duox* Reverse | 5' CAGCAGGATGTAAGGTTTCT 3' |

## γ-Irradiation of larvae

Second instar larvae were exposed to 50 Gy of γ-radiation at the rate of 2.56 Gy/min using Blood Irradiator 2000 (Board of Radiation and Isotope Technology, DAE, Mumbai). After irradiation, the larvae were transferred into media vials, maintained at 25°C for 3 hr (detection of *RPA-70GFP* and γ−H2AX) or indicated timepoints (detection of pChk1), after which they were sacrificed.

## Hydrogen peroxide treatment

Animals were dissected in PBS and flipped inside out to expose the tracheae. They were then incubated with specific concentrations of $H_2O_2$ in PBS at room temperature. For detection of 8-oxo-dG, the larvae were immediately washed in PBS and fixed with 4% PFA and immunostaining was performed as described above. For detection of pChk1, the specimens were washed in PBS immediately and ice-cold PFA was added. The samples were fixed overnight at 4°C . Immunostaining was then performed as indicated above.

## Acknowledgements

We thank Shigeo Hayashi, Eric F Wieschaus, and Anja C Nagel for fly lines. We also thank the Central Imaging and Flow Cytometry Facility (CIFF) at inStem and Fly Facility at C-CAMP for their support. Ramalingaswami Fellowship (Department of Biotechnology, Government of India, AG) and institutional funds from inStem (AK, PC) are acknowledged.

## Additional information

### Funding

| Funder | Grant reference number | Author |
|---|---|---|
| Department of Biotechnology, Ministry of Science and Technology, India | inStem Core Funds | Arjun Guha |

The funders had no role in study design, data collection and interpretation, or the decision to submit the work for publication.

## Author contributions
Amrutha Kizhedathu, Conceptualization, Investigation, Methodology, Visualization, Writing – original draft, Writing – review and editing; Piyush Chhajed, Investigation, Methodology, Visualization, Writing – original draft, Writing – review and editing; Lahari Yeramala, Investigation, Methodology, Resources; Deblina Sain Basu, Investigation, Methodology; Tina Mukherjee, Resources; Kutti R Vinothkumar, Conceptualization, Resources, Writing – original draft; Arjun Guha, Conceptualization, Funding acquisition, Methodology, Supervision, Visualization, Writing – original draft, Writing – review and editing

## Author ORCIDs
Piyush Chhajed ![ORCID] http://orcid.org/0000-0002-8705-6773
Tina Mukherjee ![ORCID] http://orcid.org/0000-0003-3776-5536
Arjun Guha ![ORCID] http://orcid.org/0000-0002-3753-1484

## Decision letter and Author response
Decision letter https://doi.org/10.7554/eLife.68636.sa1
Author response https://doi.org/10.7554/eLife.68636.sa2

# Additional files

## Supplementary files
• Transparent reporting form

## Data availability
All data generated or analysed during this study are included in the manuscript and supporting file; Source Data files have been provided for Figures 1,2,3,4.

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
