## [Decision Letter]

**Acceptance summary:**

The studies reported here represent a significant advance of the authors' prior work on the regulation of G2 arrest in larval tracheoblasts. Two well-known tumor suppressor genes, ATR and Chk1, are shown to have a new function in sensing oxidative stress agents. The identification of ROS as a developmental regulator of ATR activation is likely to be of interest to a broad audience.

**Decision letter after peer review:**

Thank you for submitting your article "Duox generated reactive oxygen species activate ATR/Chk1 to induce G2 arrest in *Drosophila* tracheoblasts" for consideration by *eLife*. Your article has been reviewed by 3 peer reviewers, including Amin S. Ghabrial as the Reviewing Editor and Reviewer #1, and the evaluation has been overseen by Didier Stainier as the Senior Editor. The following individuals involved in review of your submission have agreed to reveal their identity: Greg Beitel (Reviewer #2); Bruce A Edgar (Reviewer #3).

Essential revisions:

Results

1) A number of key results are supported by photo-micrographs, with only one example shown. Quantify results for: Figure 1B-I, Figure 2F, Figure 5B-I. Also, concerns have been raised about the significance of some results" "The significance of the cell cycle changes shown is quite weak (*p<.05; Figure 1JK, 2GH). The same is true for the Duox mRNA measurements (Figure 2A)." These experiments should have a higher n, which may raise the p value, and the authors should also consider showing all their data using dot plots with mean and error whiskers rather than bars.

Figure 1:

2) H2DCFDA and DHE staining look different in the images shown, as the former appears to be from a glancing section and the latter a section that reveals the tube lumen. For non-specialists an explanation is needed in the figure legends, or better yet, comparable sections should be shown.

3) "Levels of H2DCFDA and DHE were found to be significantly lower in Btl- SOD1 expressing animals compared to controls (Figure 1 H-I, compare with Figure 1 B, E, n ≥ 6 tracheae per condition per experiment, n = 3)."

For Figure 1H and I, the stage examined has been moved above the panels, resulting in two of three reviewers concluding that stage examined was not described. Move the L2 label to the side as in B-G and also add this information to the figure legend.

4) The diameter of the tube and the nuclei shown in H, I appear much larger than in wild type L2 (1B). Does SOD1 overexpression alter tracheal size? Scale bar in 1H appears slightly longer than in 1B, so images can't be directly compared; however, both scale bars are 10 microns and the difference in size doesn't seem to be fully accounted for by difference in magnification. It would be helpful if images, especially of the same developmental stages, are shown at the same scale.

5) Should not the control late L3 have significantly more cell numbers than the early L3 or L2 as the cells re-enter mitosis? The number of animals checked should be more (at least 10 for each condition for better statistical confidence). Since the authors see a precocious mitotic re-entry of the tracheoblasts at 16-24h L3 time point, a corresponding imaging assay for the H2DCFDA/DHE in the control versus btl-SOD1 animals would have helped support the inverse association between ROS levels and mitotic re-entry.

"Analysis of the frequencies of cells and pH3+ figures showed that SOD1 overexpression resulted in precocious cell division from 0-8 h L3 (Figure 1 J, K, n {greater than or equal to} 7 tracheae per timepoint, Figure 1 – Source data 1,2). "

For the 0-8 hr timepoint, how many PH3 nuclei are observed? If n is approximately 7 trachea, it is not clear that more than a single PH3 nucleus is observed at the 0-8 hr timepoint…

6) In Figure 1 J-L, if pChk1 activity is reduced in L2 in btl-SOD1 condition, shouldn't we expect greater cell numbers or higher pH3+ cell frequency than the control in L2 or early L3?

Figure 1, Supplement 1:

7) It is hard to evaluate the mitochondrial data since it is not stated what the ratio would be if there is no or low ROS. Authors show a dramatic decrease from L2 to L3 and state that this differs from cytoplasmic ROS dynamics based on the presence of detectable ROS in early L3 that becomes undetectable by the time that cell are released from G2 arrest. Not clear that mitochondrial data could not be consistent with the observed change in cell cycle behavior.

The authors should clarify and comment.

Does sod1 overexpression alter mitochondrial ROS?

Figure 2:

8) In Figure 2A, since the authors claim that differential effects of SOD1 starts earlier at 16-24hrs, they should show results for the Duox mRNA changes at this time point too.

9) In Figure 2B-E, exact time points for the images should be mentioned.

10) Figure 2F, there is an image duplication. The control images for Figure 1L and 2F are the same. Considering the time sensitivity of the results, each experiment should have an independent, paired control group.

11) Figure 2G, H and others. "Tr2 Cell Number" would be more informative than "Cell number"

Figure 3:

12) The results in Figure 3D suggest that reduction in ROS levels leads to precocious mitotic re-entry, independent of Chk1 mRNA level. Since low ROS is associated with low pChk1 (Figure 1L), increased Chk1 mRNA levels should not matter because there is no increased ATR activity. A better experiment to show novel ATR/Chk1 regulation by ROS would have been to increase or decrease ATR levels in btl-DuoxRNAi or btl-Duox conditions.

The listed genotype for "Btl-Duox[RNAi] Chk1" looks like it has an unspecified Chk1 mutation rather than UAS-Chk1 overexpression. Clearer alternatives include "Btl-Duox[RNAi], Btl-Chck1" or "Btl Gal4: UAS-Duox[RNAi] UAS-Chk1".

Figure 4:

13) In Figure 4E-H, the exact time range should be mentioned.

Introduction

14) "Interestingly, there is evidence that these kinases can also be activated by non-canonical mechanisms that are not dependent on DNA damage (Guo et al., 2010)" – this citation discusses just ROS, not other mechanisms. If there are other mechanisms in addition ROS, authors should include them and the appropriate citations.

15) Also, there is considerable literature that supports ROS activation of ATR, although perhaps as part of a DNA damage response. Some of these should be cited and discussed in the context of the work in this study – for example "APE2 is required for ATR-Chk1 checkpoint activation in response to oxidative stress" by Willis et al.

16) The introduction seems to ignore existing literature about the mechanisms of G2 arrest and mitotic re-entry of the tracheoblasts in the Tr2 metamere. Relevant literature (e.g. http://dx.doi.org/10.1016/j.celrep.2014.09.043) should have been discussed for a comprehensive state of the art of the field.

Discussion

17) The authors do not provide any data or discussion relevant to how Duox activity is developmentally regulated. This interesting point should be addressed, if not with experimental data, then at least with some discussion of potential mechanisms.

18) In addition, the authors should note in their conclusions and the discussion that these effects on the cell cycle are partial (i.e. the timing and amount of cell cycle re-activation is only mildly affected by Duox-RNAi or Chk-RNAi).

*Reviewer #1 (Recommendations for the authors):*

In their manuscript entitled "Duox generated reactive oxygen species activate ATR/Chk1 to induce G2 arrest in *Drosophila* tracheoblasts" Kizhedathu and colleagues follow up on their previous findings that G2 arrest of larval tracheoblasts depends on phosphorylation of Chk1 downstream of ATR activation. Previous work showed that ATR-dependent phosphorylation of Chk1 was occurring in the absence of DNA damage, raising the question of how ATR activity was controlled in this developmental context. The authors report here that G2 arrested tracheoblasts show high levels of ROS, which decrease as the cells initiate active cycling. They determine that high ROS is required for Chk1 phosphorylation and G2 arrest, and that ROS levels are dependent upon hydrogen peroxide production by Duox. The work establishes a novel mechanism for the developmental regulation of ATR-Chk1.

The authors first investigate the presence of ROS in tracheoblasts during larval life. They determine that ROS levels are elevated during stages when tracheoblasts are arrested in G2 and decline at about the point that cells re-enter the cell cycle. They demonstrate that ROS levels are relevant to G2 arrest by showing that overexpression of sod can reduce ROS and induce premature re-entry.

Authors next turn to the source of ROS generation in tracheoblasts. They examine duox as a candidate given that it produces hydrogen peroxide and was previously described to be expressed at high levels in trachea. They examine duox mRNA at different larval stages by qPCR and find that expression correlates with ROS levels. Knockdown of duox by RNAi resulted in decreased ROS, decreased pChk1 and premature re-entry.

To determine how ROS/duox regulate pChk1, authors tested whether Wnt expression, which acts on Chk1 at the transcriptional level, was affected; it was not. Likewise, ATR transcript levels were not affected. Overexpression of Chk1 could not rescue G2 arrest in the absence of ROS. Authors tested whether ROS levels were associated with DNA damage, as that would fit canonical ATR activation. Any DNA damage was below the threshold of detection. Because DNA damage-induced activation of ATR/Chk1 requires known co-factors, the authors were also able to genetically test a requirement for the DNA damage-sensing pathway in the developmental activation of ATR/CHK1. They found that knockdown of ATRIP, TOPBP1, and Claspin were able to block γ-radiation induced pChk, but not ROS induced pChk in tracheoblasts. Authors also showed that exposure to hydrogen peroxide can induce pChk in the absence of Duox, but not in the absence of ATR. The authors also show that the kinetics of Chk phosphorylation in response to ROS is much more rapid than in response to DNA damage.

Overall, this study makes a valuable advance on the prior work. I favor publication subsequent to addressing the following issues:

Introduction: "Interestingly, there is evidence that these kinases can also be activated by non-canonical mechanisms that are not dependent on DNA damage (Guo et al., 2010)" – this citation discusses just ROS, not other mechanisms. If there are other mechanisms in addition ROS, authors should include them and the appropriate citations.

Also, there is considerable literature that supports ROS activation of ATR, although perhaps as part of a DNA damage response. Some of these should be cited and discussed in the context of the work in this study – for example "APE2 is required for ATR-Chk1 checkpoint activation in response to oxidative stress" by Willis et al.

"Taken together, the analysis of ROS reporters showed that dynamics of mitochondrial and cytoplasmic ROS are different. Pertinently, these studies showed that dynamics of cytoplasmic ROS paralleled G2 arrest and mitotic re-entry." Does sod1 overexpression alter mitochondrial ROS?

H2DCFDA and DHE staining look different in the images shown, as the former appears to be from a glancing section and the latter a section that reveals the tube lumen. For non-specialists an explanation is needed in the figure legends, or alternatively, comparable sections should be shown.

"Levels of H2DCFDA and DHE were found to be significantly lower in Btl- SOD1 expressing animals compared to controls (Figure 1 H-I, compare with Figure 1 B, E, n {greater than or equal to} 6 tracheae per condition per experiment, n = 3)." Neither the text nor the figure legends specify what stage is shown in H and I. The implication is that L2 larvae are shown given the comparison to B and E; however, the diameter of the tube in H, I appears consistent with L3. Authors must specify the stage shown and account for why the tube size appears off – I note that the scale bars are 10 microns in both images and that the bar is slightly larger in H, I; this may account for some of the difference…

It is hard to evaluate the mitochondrial data since it is not stated what the ratio would be if there is no or low ROS. Authors show a dramatic decrease from L2 to L3 and state that this differs from cytoplasmic ROS dynamics based on the presence of detectable ROS in early L3 that becomes undetectable by the time that cell are released from G2 arrest. Not clear that mitochondrial data could not be consistent with the observed change in cell cycle behavior.

"Analysis of the frequencies of cells and pH3+ figures showed that SOD1 overexpression resulted in precocious cell division from 0-8 h L3 (Figure 1 J, K, n ≥ 7 tracheae per timepoint, Figure 1 – Source data 1,2)." For the 0-8 hr timepoint, how many PH3 nuclei are observed? If n is approximately 7 trachea, it is not clear that more than a single PH3 nucleus is observed at the 0-8 hr timepoint…

"the mechanism for the activation of ATR/Chk1 post mitotic re-entry is also non-canonical. " This section needs more explanation. As I understand it, the observation is that btl>chk1 RNAi cells, after precocious re-entry, begin to cycle more slowly as compared to btl>sod1 overexpressing cells. However, it is not clear to me that this reflects another requirement for ATR/pChk. So far as I am aware the authors have not shown post-re-entry pChk1 staining or made clear why they think the difference between the two isn't due to an as yet unappreciated role of ROS.

*Reviewer #2 (Recommendations for the authors):*

All figures. The authors should check *eLife* policy on use of simple bar and error whiskers in graphs ( I haven't looked at that), but the authors are encouraged to show all their data using dot plots with mean and error whiskers rather than bars.

The authors use 100 μm exogenous H_2_O_2_ to rescue the Duox phenotype. Is this a relevant level of H_2_O_2_? The authors should indicate how this compares to the normal intracellular levels of H_2_O_2_ (or equivalent reactive species), and how it compares to H_2_O_2_ levels used in ROS sensing pathways in typical experiments in this area? Some context for the reader would be very helpful.

The methods need to state that panels that show no signal were imaged using the same settings as the controls that did show an image. E.g. Figure 1B, C vs Figure 1 D,H.

For data in Figure 2, the authors are encouraged to quantify the degree of knockdown of Duox by the two RNAi lines to determine how sensitive the pathway is to Duox levels. For example, if knockdown is only 75%, this will underscore the importance of the pathway.

The blue "DAPI" label is invisible in many figures. The red pChk1 is frequently hard to read too. The authors are encouraged to use a white background behind the lettering as they did for Figure 4 B and Figure 4 Supplement 1.

P 5, paragraph 1, line 7. Please define "FUCCI".

Figure 3 D The listed genotype for "Btl-Duox[RNAi] Chk1" looks like it has an unspecified Chk1 mutation rather than UAS-Chk1 overexpression. Clearer alternatives include "Btl-Duox[RNAi], Btl-Chck1" or "Btl Gal4: UAS-Duox[RNAi] UAS-Chk1".

In several cases, it would be helpful for the authors to better label their figures so the reader didn't have to dig into the legend to understand the figure. Examples:

Figure 1, Supplement 1 "Ratio of 405/458 for MTS roGFP2" would be more informative than "Ratio of 405/458". What dye or fluorescent marker is this?

Figure 2G, H and others. "Tr2 Cell Number" would be more informative than "Cell number"

*Reviewer #3 (Recommendations for the authors):*

This paper is a continuation of two related works on the same topic, from the same authors, also published in e*Life* (2018, 2020). The previous papers show the roles of ATR and WNT, but do not address ROS or Duox. While this new paper is generally convincing, it has some technical and presentational weaknesses that should be addressed in order to bring it up to accepted standards for this experimental system and field. These are described below.

1. A number of key results are supported by photo-micrographs, with only one example shown. These results should be quantified from multiple samples, and the data presented in graphical form with standard deviations and p values (in addition to the pictorial examples, which are quite good). Experiments that should quantified include: (1) Figure 1B-I (H2DCFDA, DHE); (2) Figure 2F (pChk1); (3) Figure 5B-I (pChk1).

2. The significance of the cell cycle changes shown is quite weak (*p<.05; Figure 1JK, 2GH). The same is true for the Duox mRNA measurements (Figure 2A). These experiments should be repeated so as to have enough samples to derive p values of <.001. In addition, the authors should note in their conclusions and the discussion that these effects on the cell cycle are partial (i.e. the timing and amount of cell cycle re-activation is only mildly affected by Duox-RNAi or Chk-RNAi).

3. The authors do not provide any data or discussion relevant to how Duox activity is developmentally regulated. This interesting point should be addressed, if not with experimental data, then at least with some discussion of potential mechanisms.

4. The introduction seems to ignore existing literature about the mechanisms of G2 arrest and mitotic re-entry of the tracheoblasts in the Tr2 metamere. Relevant literature (e.g. http://dx.doi.org/10.1016/j.celrep.2014.09.043) should have been discussed for a comprehensive state of the art of the field.

5. In Figure 1 H, it is not clear at which stage the image was taken.

6. In Figure 1J, the numbers for WL3 stage do not really add any value in this graph. The WL3 numbers skew the graph and obscure the differences in the earlier stages, which the authors emphasize. Should not the control late L3 have significantly more cell numbers than the early L3 or L2 as the cells re-enter mitosis? The number of animals checked should be more (at least 10 for each condition for better statistical confidence). Since the authors see a precocious mitotic re-entry of the tracheoblasts at 16-24h L3 time point, a corresponding imaging assay for the H2DCFDA/DHE in the control versus btl-SOD1 animals would have helped support the inverse association between ROS levels and mitotic re-entry.

7. In Figure 1 L, if pChk1 activity is reduced in L2 in btl-SOD1 condition, shouldn't we expect greater cell numbers or higher pH3+ cell frequency than the control in L2 or early L3?

8. In Figure 2A, since the authors claim that differential effects of SOD1 starts earlier at 16-24hrs, they should show results for the Duox mRNA changes at this time point too.

9. In Figure 2B-E, exact time points for the images should be mentioned.

10. Figure 2F, there is an image duplication. The control images for Figure 1L and 2F are the same. Considering the time sensitivity of the results, each experiment should have an independent, paired control group.

11. The results in Figure 3D suggest that reduction in ROS levels leads to precocious mitotic re-entry, independent of Chk1 mRNA level. Since low ROS is associated with low pChk1 (Figure 1L), increased Chk1 mRNA levels should not matter because there is no increased ATR activity. A better experiment to show novel ATR/Chk1 regulation by ROS would have been to increase or decrease ATR levels in btl-DuoxRNAi or btl-Duox conditions.

12. In Figure 4E-H, the exact time range should be mentioned.

13. In Figure 5J, the model should reflect the fact the authors have not yet identified a direct mechanism for ROS-mediated ATR/Chk1 regulation. Use of a dotted line would be better. It could also be helpful to speculate about the developmental regulation of Duox (and Wnt) here in the figure, if the authors have a suggested mechanism for their regulation.

---

## [Author Response]

Essential revisions:Results1) A number of key results are supported by photo-micrographs, with only one example shown. Quantify results for: Figure 1B-I, Figure 2F, Figure 5B-I.

We have quantified the results for Figure 1 B-I and the data is provided in Figure 1 Figure supplement 1. We have also quantified the pChk1 staining in Figure 2F, Figure 5B-I and the data has been included in Figure 1 Figure supplement 2.

Also, concerns have been raised about the significance of some results" "The significance of the cell cycle changes shown is quite weak (*p<.05; Figure 1JK, 2GH). The same is true for the Duox mRNA measurements (Figure 2A)." These experiments should have a higher n, which may raise the p value, and the authors should also consider showing all their data using dot plots with mean and error whiskers rather than bars.

The exact p values for the datasets have been added to the source data files associated with the figures. The p value across experiments is typically less than 0.01.

We have now modified the figures to represent the data previously shown in bar graphs as box-and-whiskers plots. The raw values associated with these figures are included in the corresponding source data files.

Figure 1:2) H2DCFDA and DHE staining look different in the images shown, as the former appears to be from a glancing section and the latter a section that reveals the tube lumen. For non-specialists an explanation is needed in the figure legends, or better yet, comparable sections should be shown.

We have revised the figure to show comparable sections.

3) "Levels of H2DCFDA and DHE were found to be significantly lower in Btl- SOD1 expressing animals compared to controls (Figure 1 H-I, compare with Figure 1 B, E, n ≥ 6 tracheae per condition per experiment, n = 3)."For Figure 1H and I, the stage examined has been moved above the panels, resulting in two of three reviewers concluding that stage examined was not described. Move the L2 label to the side as in B-G and also add this information to the figure legend.

We have revised the labels as suggested.

4) The diameter of the tube and the nuclei shown in H, I appear much larger than in wild type L2 (1B). Does SOD1 overexpression alter tracheal size? Scale bar in 1H appears slightly longer than in 1B, so images can't be directly compared; however, both scale bars are 10 microns and the difference in size doesn't seem to be fully accounted for by difference in magnification. It would be helpful if images, especially of the same developmental stages, are shown at the same scale.

We have revised the figure and have shown all the images at the same scale.

5) Should not the control late L3 have significantly more cell numbers than the early L3 or L2 as the cells re-enter mitosis? The number of animals checked should be more (at least 10 for each condition for better statistical confidence). Since the authors see a precocious mitotic re-entry of the tracheoblasts at 16-24h L3 time point, a corresponding imaging assay for the H2DCFDA/DHE in the control versus btl-SOD1 animals would have helped support the inverse association between ROS levels and mitotic re-entry.

The numbers of cells in Tr2 at 32-40h L3 (25.8) is greater than the numbers of cells at early L3 (0-8h L3, 19). This difference is statistically significant (p value=0.000000062). We have now included data on ROS levels at 16-24h L3 in control (Figure 1 D, H) and btl-Sod1 (Figure 1 Figure supplement 1 C, D) animals.

"Analysis of the frequencies of cells and pH3+ figures showed that SOD1 overexpression resulted in precocious cell division from 0-8 h L3 (Figure 1 J, K, n {greater than or equal to} 7 tracheae per timepoint, Figure 1 – Source data 1,2). "For the 0-8 hr timepoint, how many PH3 nuclei are observed? If n is approximately 7 trachea, it is not clear that more than a single PH3 nucleus is observed at the 0-8 hr timepoint…

We have observed 234 nuclei out of which only two were PH3 positive.

6) In Figure 1 J-L, if pChk1 activity is reduced in L2 in btl-SOD1 condition, shouldn't we expect greater cell numbers or higher pH3+ cell frequency than the control in L2 or early L3?

We do not observe an increase in the numbers of cell or the frequencies of pH3+ cells in btl-Sod1 at L2 because of an additional mechanism that actuates arrest G2 in L2 (Kizhedathu et al., 2018). We do observe a subtle increase in the frequencies of pH3+ cells in both *btl-SOD1* (Figure 1M) and *btl-Duox^RNAi^* (Figure 2H) animals at 0-8 h L3.

Figure 1, Supplement 1:7) It is hard to evaluate the mitochondrial data since it is not stated what the ratio would be if there is no or low ROS. Authors show a dramatic decrease from L2 to L3 and state that this differs from cytoplasmic ROS dynamics based on the presence of detectable ROS in early L3 that becomes undetectable by the time that cell are released from G2 arrest. Not clear that mitochondrial data could not be consistent with the observed change in cell cycle behavior.The authors should clarify and comment.Does sod1 overexpression alter mitochondrial ROS?

We accept that this data in not central to our narrative and it has been removed in the revised version.

Figure 2:8) In Figure 2A, since the authors claim that differential effects of SOD1 starts earlier at 16-24hrs, they should show results for the Duox mRNA changes at this time point too.

We have quantified Duox mRNA at 16-24h L3 and the data has been added to Figure 2A.

9) In Figure 2B-E, exact time points for the images should be mentioned.

Experiments in Figure 2B-E were performed on L2 animals. We have specified the time points in Figure 2B-E.

10) Figure 2F, there is an image duplication. The control images for Figure 1L and 2F are the same. Considering the time sensitivity of the results, each experiment should have an independent, paired control group.

We have now replaced the image for control in Figure 2F with another image of control trachea from the same experiment.

11) Figure 2G, H and others. "Tr2 Cell Number" would be more informative than "Cell number"

We have changed the labels accordingly.

Figure 3:12) The results in Figure 3D suggest that reduction in ROS levels leads to precocious mitotic re-entry, independent of Chk1 mRNA level. Since low ROS is associated with low pChk1 (Figure 1L), increased Chk1 mRNA levels should not matter because there is no increased ATR activity. A better experiment to show novel ATR/Chk1 regulation by ROS would have been to increase or decrease ATR levels in btl-DuoxRNAi or btl-Duox conditions.

We have overexpressed ATR in *btl-Duox^RNAi^* animals and counted the number of cells in Tr2 DT. We found that the numbers were higher than control and comparable to *btl-Duox^RNAi^* animals (Figure 3D). This show that ATR overexpression is unable to rescue the Duox phenotype that reduced expression of ATR is not the reason underlying the phenotype.

The listed genotype for "Btl-Duox[RNAi] Chk1" looks like it has an unspecified Chk1 mutation rather than UAS-Chk1 overexpression. Clearer alternatives include "Btl-Duox[RNAi], Btl-Chck1" or "Btl Gal4: UAS-Duox[RNAi] UAS-Chk1".

We have now detailed the genotype in the text to avoid any confusion.

Figure 4:13) In Figure 4E-H, the exact time range should be mentioned.

We have revised the figure and specified that the experiments described in Figure 4 E-H were performed at L2 stage.

Introduction14) "Interestingly, there is evidence that these kinases can also be activated by non-canonical mechanisms that are not dependent on DNA damage (Guo et al., 2010)" – this citation discusses just ROS, not other mechanisms. If there are other mechanisms in addition ROS, authors should include them and the appropriate citations.

A pioneering study that demonstrates ATR activation in response to mechanical stress (Kumar et al., 2014), in the absence of DNA damage, has been mentioned in the Introduction and the Discussion.

15) Also, there is considerable literature that supports ROS activation of ATR, although perhaps as part of a DNA damage response. Some of these should be cited and discussed in the context of the work in this study – for example "APE2 is required for ATR-Chk1 checkpoint activation in response to oxidative stress" by Willis et al.

The study has been mentioned and cited in the Introduction.

16) The introduction seems to ignore existing literature about the mechanisms of G2 arrest and mitotic re-entry of the tracheoblasts in the Tr2 metamere. Relevant literature (e.g. http://dx.doi.org/10.1016/j.celrep.2014.09.043) should have been discussed for a comprehensive state of the art of the field.

Done.

Discussion17) The authors do not provide any data or discussion relevant to how Duox activity is developmentally regulated. This interesting point should be addressed, if not with experimental data, then at least with some discussion of potential mechanisms.

We have now added the following paragraph in the discussion

How are ROS levels regulated during development? Analysis of Duox mRNA levels in tracheoblasts shows that expression is high in L2 and early L3 and drops significantly at 32-40 h L3. This parallels the expression of Chk1. We have previously shown that the levels of Chk1 mRNA are regulated transcriptionally by the Wnt signallng pathway. We determined whether Wnt signalling also regulates Duox expression and ROS levels. This analysis showed that ROS levels are unaffected in Wnt pathway mutants (data not shown). We infer that Wnt signalling pathway does not regulate Duox nor ROS levels. Juvenile hormone (JH) has been shown to act as a negative regulator of cell proliferation in tracheae (Djabrayan and Casanova, 2016). Pertinently, the levels of JH are high in L2 and early L3 and drop mid L3 (Dubrovsky, 2005). Thus, the timecourse of JH levels parallel the timecourses of both Duox and Chk1 expression. Future experiments will test the possibility that JH signaling in trachea regulates Duox expression and potentially Wnt signaling/Chk1 expression as well.

18) In addition, the authors should note in their conclusions and the discussion that these effects on the cell cycle are partial (i.e. the timing and amount of cell cycle re-activation is only mildly affected by Duox-RNAi or Chk-RNAi).

Please see response to (6).

Reviewer #1 (Recommendations for the authors):[…] Overall, this study makes a valuable advance on the prior work. I favor publication subsequent to addressing the following issues:Introduction: "Interestingly, there is evidence that these kinases can also be activated by non-canonical mechanisms that are not dependent on DNA damage (Guo et al., 2010)" – this citation discusses just ROS, not other mechanisms. If there are other mechanisms in addition ROS, authors should include them and the appropriate citations.

Done,

Also, there is considerable literature that supports ROS activation of ATR, although perhaps as part of a DNA damage response. Some of these should be cited and discussed in the context of the work in this study – for example "APE2 is required for ATR-Chk1 checkpoint activation in response to oxidative stress" by Willis et al.

Please see response to Essential revisions/Major concerns: (15)

"Taken together, the analysis of ROS reporters showed that dynamics of mitochondrial and cytoplasmic ROS are different. Pertinently, these studies showed that dynamics of cytoplasmic ROS paralleled G2 arrest and mitotic re-entry." Does sod1 overexpression alter mitochondrial ROS?

Please see response to Essential revisions/Major concerns: (7)

H2DCFDA and DHE staining look different in the images shown, as the former appears to be from a glancing section and the latter a section that reveals the tube lumen. For non-specialists an explanation is needed in the figure legends, or alternatively, comparable sections should be shown.

Please see response to Essential revisions/Major concerns (2)

"Levels of H2DCFDA and DHE were found to be significantly lower in Btl- SOD1 expressing animals compared to controls (Figure 1 H-I, compare with Figure 1 B, E, n {greater than or equal to} 6 tracheae per condition per experiment, n = 3)." Neither the text nor the figure legends specify what stage is shown in H and I. The implication is that L2 larvae are shown given the comparison to B and E; however, the diameter of the tube in H, I appears consistent with L3. Authors must specify the stage shown and account for why the tube size appears off – I note that the scale bars are 10 microns in both images and that the bar is slightly larger in H, I; this may account for some of the difference…

Please see response to Essential revisions/Major concerns: (3), (4)

It is hard to evaluate the mitochondrial data since it is not stated what the ratio would be if there is no or low ROS. Authors show a dramatic decrease from L2 to L3 and state that this differs from cytoplasmic ROS dynamics based on the presence of detectable ROS in early L3 that becomes undetectable by the time that cell are released from G2 arrest. Not clear that mitochondrial data could not be consistent with the observed change in cell cycle behavior.

Please see response to Essential revisions/Major concerns: (7)

"Analysis of the frequencies of cells and pH3+ figures showed that SOD1 overexpression resulted in precocious cell division from 0-8 h L3 (Figure 1 J, K, n {greater than or equal to} 7 tracheae per timepoint, Figure 1 – Source data 1,2). " For the 0-8 hr timepoint, how many PH3 nuclei are observed? If n is approximately 7 trachea, it is not clear that more than a single PH3 nucleus is observed at the 0-8 hr timepoint…

Please see response to Essential revisions/Major concerns: (5)

"the mechanism for the activation of ATR/Chk1 post mitotic re-entry is also non-canonical." This section needs more explanation. As I understand it, the observation is that btl>chk1 RNAi cells, after precocious re-entry, begin to cycle more slowly as compared to btl>sod1 overexpressing cells. However, it is not clear to me that this reflects another requirement for ATR/pChk. So far as I am aware the authors have not shown post-re-entry pChk1 staining or made clear why they think the difference between the two isn't due to an as yet unappreciated role of ROS.

We have shown previously that there are two roles for ATR/Chk1 in Tr2 tracheoblasts (Kizhedathu et al., 2018). First, to arrest cells in G2 Second, for mitoses post mitotic re-entry. We showed that these two roles can be uncoupled by inactivating Chk1 at different times (Kizhedathu et al., 2018). Since the overexpression SOD1 inhibits ATR/Chk1 activation and abrogates G2 arrest but does not delay mitoses, we infer that the mechanisms for the activation of ATR/Chk1 while the cells are arrested and post mitotic re-entry are different from each other. Although the SOD1 overexpression suggests that ROS may not be driving activation post mitotic re-entry, this is not clear yet.

This has been clarified in the Discussion.

Reviewer #2 (Recommendations for the authors):All figures. The authors should check eLife policy on use of simple bar and error whiskers in graphs ( I haven't looked at that), but the authors are encouraged to show all their data using dot plots with mean and error whiskers rather than bars.

Please see response to Essential revisions/Major concerns: (1)

The authors use 100 μm exogenous H_2_O_2_ to rescue the Duox phenotype. Is this a relevant level of H_2_O_2_? The authors should indicate how this compares to the normal intracellular levels of H_2_O_2_ (or equivalent reactive species), and how it compares to H_2_O_2_ levels used in ROS sensing pathways in typical experiments in this area? Some context for the reader would be very helpful.

The range of H_2_O_2_ concentrations we have used in study have been defined by (1) experiments done on cell lines to induce DNA damage (1 mM, Fujii et al., 2002) and (2) experiments done on cell lines (25 µm-250 µm) and purified proteins (62.5 µm-22mM) to activate ATM (Guo et al., 2010). The concentration of H_2_O_2_ used in the *btl-Duox^RNAi^* rescue is a concentration at which we do not detect any DNA damage and a concentration at which ATM can be activated.

The methods need to state that panels that show no signal were imaged using the same settings as the controls that did show an image. E.g. Figure 1B, C vs Figure 1 D,H.

Done.

For data in Figure 2, the authors are encouraged to quantify the degree of knockdown of Duox by the two RNAi lines to determine how sensitive the pathway is to Duox levels. For example, if knockdown is only 75%, this will underscore the importance of the pathway.

We have quantified the mRNA levels of Duox in *btl-Duox^RNAi^* tracheae at L2. The mRNA levels were undetectable

The blue "DAPI" label is invisible in many figures. The red pChk1 is frequently hard to read too. The authors are encouraged to use a white background behind the lettering as they did for Figure 4 B and Figure 4 Supplement 1.

Done.

P 5, paragraph 1, line 7. Please define "FUCCI".

Done.

Figure 3 D The listed genotype for "Btl-Duox[RNAi] Chk1" looks like it has an unspecified Chk1 mutation rather than UAS-Chk1 overexpression. Clearer alternatives include "Btl-Duox[RNAi], Btl-Chck1" or "Btl Gal4: UAS-Duox[RNAi] UAS-Chk1".

Please see response to Essential revisions/Major concerns: (12)

In several cases, it would be helpful for the authors to better label their figures so the reader didn't have to dig into the legend to understand the figure. Examples:Figure 1, Supplement 1 "Ratio of 405/458 for MTS roGFP2" would be more informative than "Ratio of 405/458". What dye or fluorescent marker is this?

Please see response to Essential revisions/Major concerns: (7)

Figure 2G, H and others. "Tr2 Cell Number" would be more informative than "Cell number"

Done.

Reviewer #3 (Recommendations for the authors):This paper is a continuation of two related works on the same topic, from the same authors, also published in eLife (2018, 2020). The previous papers show the roles of ATR and WNT, but do not address ROS or Duox. While this new paper is generally convincing, it has some technical and presentational weaknesses that should be addressed in order to bring it up to accepted standards for this experimental system and field. These are described below.1. A number of key results are supported by photo-micrographs, with only one example shown. These results should be quantified from multiple samples, and the data presented in graphical form with standard deviations and p values (in addition to the pictorial examples, which are quite good). Experiments that should quantified include: (1) Figure 1B-I (H2DCFDA, DHE); (2) Figure 2F (pChk1); (3) Figure 5B-I (pChk1).

Please see response to Essential revisions/Major concerns: (11)

2. The significance of the cell cycle changes shown is quite weak (*p<.05; Figure 1JK, 2GH). The same is true for the Duox mRNA measurements (Figure 2A). These experiments should be repeated so as to have enough samples to derive p values of <.001. In addition, the authors should note in their conclusions and the discussion that these effects on the cell cycle are partial (i.e. the timing and amount of cell cycle re-activation is only mildly affected by Duox-RNAi or Chk-RNAi).

Please see response to Essential revisions/Major concerns: (1)

3. The authors do not provide any data or discussion relevant to how Duox activity is developmentally regulated. This interesting point should be addressed, if not with experimental data, then at least with some discussion of potential mechanisms.

Please see response to Essential revisions/Major concerns: (18)

4. The introduction seems to ignore existing literature about the mechanisms of G2 arrest and mitotic re-entry of the tracheoblasts in the Tr2 metamere. Relevant literature (e.g. http://dx.doi.org/10.1016/j.celrep.2014.09.043) should have been discussed for a comprehensive state of the art of the field.

Please see response to Essential revisions/Major concerns: (16)

5. In Figure 1 H, it is not clear at which stage the image was taken.

Please see response to Essential revisions/Major concerns: (3)

6. In Figure 1J, the numbers for WL3 stage do not really add any value in this graph. The WL3 numbers skew the graph and obscure the differences in the earlier stages, which the authors emphasize. Should not the control late L3 have significantly more cell numbers than the early L3 or L2 as the cells re-enter mitosis? The number of animals checked should be more (at least 10 for each condition for better statistical confidence). Since the authors see a precocious mitotic re-entry of the tracheoblasts at 16-24h L3 time point, a corresponding imaging assay for the H2DCFDA/DHE in the control versus btl-SOD1 animals would have helped support the inverse association between ROS levels and mitotic re-entry.

Please see response to Essential revisions/Major concerns: (5)

7. In Figure 1 L, if pChk1 activity is reduced in L2 in btl-SOD1 condition, shouldn't we expect greater cell numbers or higher pH3+ cell frequency than the control in L2 or early L3?

Please see response to Essential revisions/Major concerns: (6)

8. In Figure 2A, since the authors claim that differential effects of SOD1 starts earlier at 16-24hrs, they should show results for the Duox mRNA changes at this time point too.

Please see response to Essential revisions/Major concerns: (8)

9. In Figure 2B-E, exact time points for the images should be mentioned.

Done.

10. Figure 2F, there is an image duplication. The control images for Figure 1L and 2F are the same. Considering the time sensitivity of the results, each experiment should have an independent, paired control group.

Please see response to Essential revisions/Major concerns: (10)

11. The results in Figure 3D suggest that reduction in ROS levels leads to precocious mitotic re-entry, independent of Chk1 mRNA level. Since low ROS is associated with low pChk1 (Figure 1L), increased Chk1 mRNA levels should not matter because there is no increased ATR activity. A better experiment to show novel ATR/Chk1 regulation by ROS would have been to increase or decrease ATR levels in btl-DuoxRNAi or btl-Duox conditions.

Please see response to Essential revisions/Major concerns: (12)

12. In Figure 4E-H, the exact time range should be mentioned.

Done.

13. In Figure 5J, the model should reflect the fact the authors have not yet identified a direct mechanism for ROS-mediated ATR/Chk1 regulation. Use of a dotted line would be better. It could also be helpful to speculate about the developmental regulation of Duox (and Wnt) here in the figure, if the authors have a suggested mechanism for their regulation.

We have modified the figure and used a dotted line to indicate the mechanism of ROS based regulation of ATR/Chk1 in Figure 5 J.